# PRE-TRAINING ROBO-CENTRIC WORLD MODELS FOR EFFICIENT VISUAL CONTROL

## ABSTRACT

Humans can accurately anticipate their movements to behave as expected in various manipulation tasks. We are inspired to propose that integrating prior knowledge of robot dynamics into world models can effectively improve the sample efficiency of model-based reinforcement learning (MBRL) in visual robot control tasks. In this paper, we introduce the Robo-Centric World Model (RCWM), which explicitly decouples the robot dynamics from the environment and enables pre-training to learn generalized and robust robot dynamics as prior knowledge to accelerate learning new tasks. Specifically, we construct respective dynamics models for the robot and the environment and learn their interactions through cross-attention mechanism. With the mask-guided reconfiguration mechanism, we only need a few prior robot segmentation masks to guide the RCWM to disentangle the robot and environment features and learn their respective dynamics. Our approach enables independent inference of robot dynamics from the environment, allowing accurate prediction of robot movement across various unseen tasks without being distracted by environmental variations. Our results in Meta-world demonstrate that RCWM is able to efficiently learn robot dynamics, improving sample efficiency for downstream tasks and enhancing policy robustness against environmental disturbances compared to the vanilla world model in DreamerV3. Code and visualizations are available on the project website: https://robo-centric-wm.github.io.

## 1 INTRODUCTION

Model-based reinforcement learning (MBRL) holds significant promise for achieving sample-efficient learning in visual robot control tasks (Ebert et al., 2018; Hafner et al., 2019; 2020a; Sekar et al., 2020; Mendonca et al., 2021; Rybkin et al., 2021; Seo et al., 2023). By constructing world models (Ha & Schmidhuber, 2018) that approximate the dynamics of the real environment, agents can leverage generated imaginary trajectories for planning (Hafner et al., 2019; Ye et al., 2021; Hansen et al., 2022; Chung et al., 2023; Hansen et al., 2024) or behavioral learning (Hafner et al., 2020b; 2023; Seo et al., 2023), thereby reducing the interaction with the environment. However, learning accurate world models from scratch, especially in the absence of prior knowledge, poses a formidable challenge. The presence of model errors hinders the policy learning, restricting the further improvement of sample efficiency in MBRL.

One feasible way to address this challenge is to pre-train world models to obtain useful information for downstream tasks to accelerate learning. Inspired by the success of pre-training paradigms in domains such as computer vision (CV) (Deng et al., 2009; Radford et al., 2021) and natural language processing (NLP) (Brown et al., 2020; OpenAI, 2022), recent works have aimed to investigate effective pre-training methods for world models (Seo et al., 2022; Wu et al., 2023; Mazzaglia et al., 2023; Rajeswar et al., 2023). For visual manipulation tasks that require precise estimation of the current robot state, it is necessary for the world model to accurately capture the robot dynamics to generate realistic imaginary trajectories in response to the action for policy learning. However, existing works rarely recognize this and typically focuses on pre-training effective representations.

When humans interact with physical world, they typically have the ability to anticipate their own movements, allowing them to act aligned with their expectations (Miall & Wolpert, 1996). With this in mind, we naturally ask the following question:

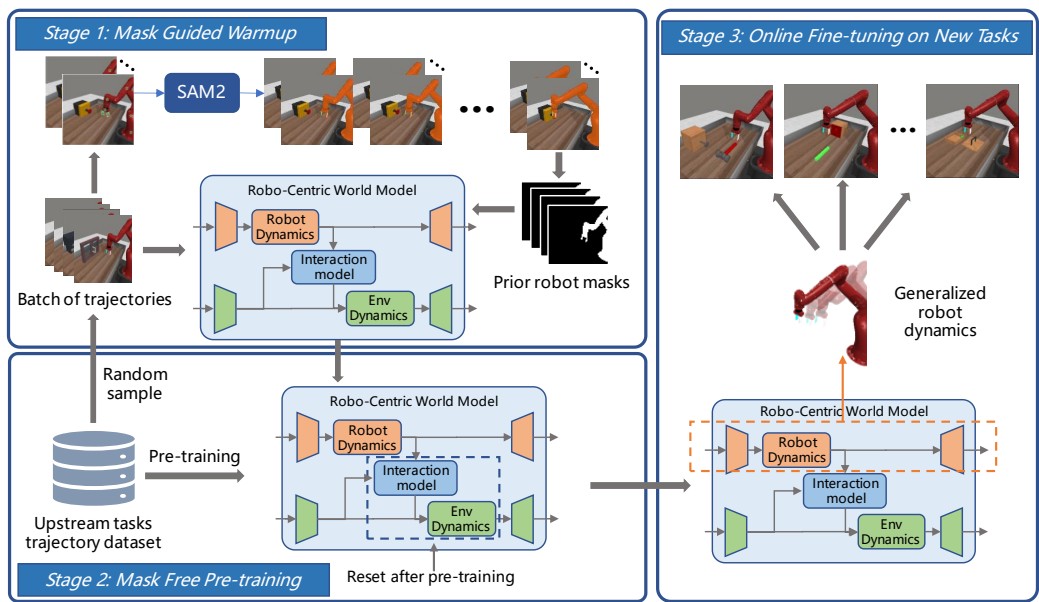

Figure 1: Overall pipeline of our approach. We first guide the warmup of the RCWM using a few trajectories with the prior robot mask generated with SAM2 to ensure that the robot branch learns the relevant features. Then we pre-train the robo-centric world model using trajectories collected on upstream tasks without prior robot masks to learn robot dynamics generalized to other tasks. Finally, we use the pre-trained world model with robot dynamics prior for learning various new tasks to improve sample efficiency.

*Can we integrate prior knowledge of robot dynamics into world models to mitigate the disparity between imagined and actual behaviors for improving sample efficiency?*

However, existing methods often rely on a single model to learn world dynamics, which results in the entanglement of robot and environment dynamics, making it difficult to provide accurate robot state estimation when confronted with new tasks. In this paper, we introduce the Robot-Centric World Model (RCWM), which is designed to extract robot dynamics through pre-training and apply this prior knowledge on new tasks to provide accurate dynamics prediction, as shown in Figure 1. Specifically, we construct two branches to learn the dynamics of the environment and robot, respectively, and use an interaction model with cross-attention mechanism to learn robot-object interaction relations. In order for both branches to capture their respective relevant features, we design the mask-guided reconstruction mechanism and use only a few prior robot segmentation masks generated by Segment-Anything-Model-2 (SAM 2) (Ravi et al., 2024) to achieve implicit feature disentanglement at the beginning of the pre-training period.

RCWM offers several advantages: (1) Accurate prediction of robot dynamics. We find that explicitly modeling robot dynamics individually results in more accurate predictions compared to learning global dynamics with a single model; (2) Robust against environmental disturbance. Due to implicit feature disentanglement, the robot branch is hardly affected by environmental disturbances, providing robust robot representation for the policy; (3) Utilization of prior masks. The architecture of RCWM naturally introduces the use of robot segmentation masks, which can enhance the prediction accuracy of robot dynamics. We evaluate and analyze RCWM on Meta-world and show that RCWM is more suitable for robots to learn a variety of manipulation tasks than the vanilla world model in DreamerV3 (Hafner et al., 2023) that uses a single dynamic model to learn global dynamics.

## 2 RELATED WORKS

**MBRL from pixels** Model-based approaches have shown promise in efficiently tackling decision problems with visual inputs, such as video games (Ha & Schmidhuber, 2018; Schrittwieser et al.,

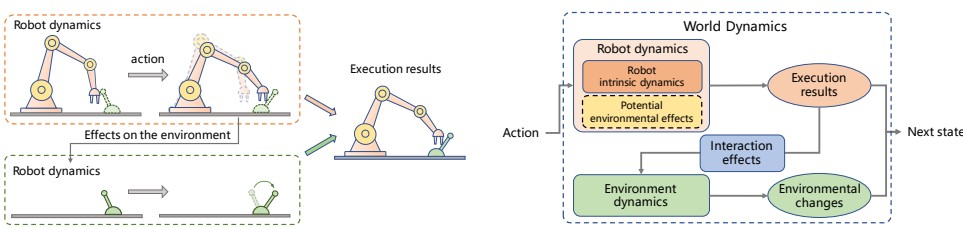

(a) Illustration of robot dynamic decoupling      (b) Simplified interaction framework

Figure 2: (a) The robot responds to action signals, driving changes in the environment as it moves. The dashed outline of the lever switch indicates the potential effect of the environment on the robot. (b) We implicitly model the potential environmental effects on the robot, represented by dashed boxes, and explicitly model the rest.

2020; Kaiser et al., 2020; Hafner et al., 2020b; Ye et al., 2021) and visual robot control tasks (Ebert et al., 2018; Hafner et al., 2020a; Sekar et al., 2020; Rybkin et al., 2021; Seo et al., 2023). Some approaches concentrate on acquiring effective representations to construct learning-friendly latent spaces, achieved by image reconstruction (Hafner et al., 2019; Seo et al., 2023) or contrastive learning (Okada & Taniguchi, 2021; Deng et al., 2022). Several approaches improve the structure of world models to achieve more accurate predictions, such as utilizing the transformer (Chen et al., 2022; Micheli et al., 2023; Zhang et al., 2023) or state space model (Deng et al., 2023; Samsami et al., 2024). In addition, there are methods that focus on extracting task-relevant information from noisy observations (Pan et al., 2022; Fu et al., 2021). Unlike existing approaches that typically learn a whole-world dynamic, we separately learn the robot dynamics and the environment dynamics, as well as the interactions between them.

**Pretraining world models** World models have recently received a lot of attention in various fields such as autonomous driving (Wang et al., 2023b;a) and video generation (Bruce et al., 2024; Zhen et al., 2024; OpenAI, 2024; Yang et al., 2024). In contrast to the CV and NLP domains, research on pre-training methods in MBRL remains relatively nascent. Some approaches learn representations that can be used for various downstream tasks by pre-training video prediction models with easily accessible action-free videos (Seo et al., 2022; Mendonca et al., 2023; Wu et al., 2023). However, these methods are of limited help when confronted with robot manipulation tasks that require accurate predictions. There are also methods that utilize unsupervised learning for task-agnostic exploration to collect data to pre-training world models (Mazzaglia et al., 2023; Rajeswar et al., 2023). Nevertheless, these methods usually require pre-training and fine-tuning in the same task scenario. Our approach provides a new idea for pre-training by integrating prior knowledge of robot dynamics into world models, and applying the extracted robot dynamics to various downstream tasks for accurate robot motion predictions.

## 3 BACKGROUND

**Problem formulation** In this paper, we focus on visual robot control tasks that we formulate as a partially observable Markov decision process (POMDP) (Sutton & Barto, 2018) represented by the tuple $(\mathcal{O}, \mathcal{A}, p, r, \gamma)$. $\mathcal{O}$ is the visual observation space, $\mathcal{A}$ is the action space, $p(o_t \mid o_{<t}, a_{<t})$ is the transition dynamics, $r$ is the reward function that maps previous observations and actions to a reward $r_t = r(o_{\leq t}, a_{<t})$, and $\gamma \in [0, 1)$ is the discount factor.

**DreamerV3** DreamerV3 (Hafner et al., 2023) is a model-based RL method that formulates the world model with a Recurrent State Space Model (RSSM) (Hafner et al., 2019), which consists of the following components:

$$
\begin{array}{llll}
\text{Representation model:} & z_t \sim q_\phi(z_t \mid h_t, o_t) & \text{Image decoder:} & \hat{o}_t \sim p_\phi(\hat{o}_t \mid s_t) \\
\text{Sequence model:} & h_t \sim f_\phi(h_{t-1}, z_{t-1}, a_{t-1}) & \text{Reward predictor:} & \hat{r}_t \sim p_\phi(\hat{r}_t \mid s_t) \\
\text{Dynamics predictor:} & \hat{z}_t \sim p_\phi(\hat{z}_t \mid h_t) & \text{Continue predictor:} & \hat{c}_t \sim p_\phi(\hat{c}_t \mid s_t)
\end{array} \quad (1)
$$

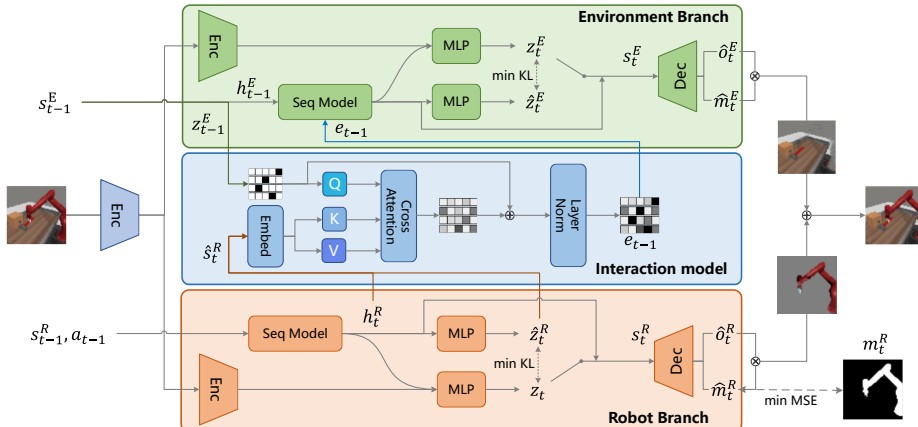

Figure 3: The architectural design of the robo-centric world model. We build separate branches for robot and environment, and design an interaction model to learn how the robot interacts with the environment. The two branches reconstruct the observations of the robot and the environment through mask-guided decoders and add them together to obtain the final reconstruction result.

The representation model embeds the current observation $o_t$ and the recurrent state $h_t$ into a low-dimensional latent space. The sequence model updates the recurrent state $h_t$ based on stochastic representation $z_{t-1}$ and action $a_{t-1}$, without direct access to $o_t$. The dynamics predictor predicts the prior stochastic representation $\hat{z}_t$ using $h_t$. The image decoder reconstructs the $o_t$ to provide supervised signals for representation learning. Additionally, the reward predictor and continue predictor estimate the reward and episode continuation flags $c_t \in \{0, 1\}$ respectively for the latent state $s_t \doteq \{h_t, z_t\}$ instead of the reconstructed observation. The overall models are jointly learned by minimizing the negative variational lower bound (ELBO)(Kingma & Welling, 2014):

$$\mathcal{L}_{KL}(q, p) \doteq \beta_{dyn} \max(1, \text{KL}\left[\text{sg}(q) \,\|\, p\right]) + \beta_{rep} \max(1, \text{KL}\left[(q) \,\|\, \text{sg}(p)\right]) \tag{2}$$

$$\mathcal{L}(\phi) \doteq \mathbb{E}_{q_\phi(z_{1:T}|a_{1:T}, o_{1:T})} \Big[ \sum_{t=1}^{T} \Big( -\ln p_\phi(o_t \,|\, s_t) - \ln p_\phi(r_t \,|\, s_t) - \ln p_\phi(c_t \,|\, s_t) \tag{3}$$
$$+ \mathcal{L}_{KL}(q_\phi(z_t \,|\, h_t, o_t), p_\phi(\hat{z}_t \,|\, h_t)) \Big) \Big].$$

For behavior learning, DreamerV3 utilizes imaginary trajectories generated by interacting with the learned world model for actor-critic learning (see Appendix A.1 for the details).

## 4 METHODS

### 4.1 ROBOT DYNAMICS DECOUPLING

We consider visual robot control scenarios in which the same robot is employed for various manipulation tasks. Although MBRL approaches can efficiently learn visual control policies, training the world model from scratch for each task still requires a significant amount of interaction with the environment, which limits further improvements in sample efficiency. Intuitively, the intrinsic dynamics of the robot remain consistent across different tasks. However, existing approaches typically use a single model to learn the world dynamics which couples the robot dynamics with the environment dynamics. As a result, when confronted with new tasks, the world model is unable to avoid the visual interference on robot dynamics prediction caused by environmental changes, thus making it difficult to directly provide generalized prior knowledge about the robot dynamics for learning new tasks. If we can individually extract this prior knowledge from the collected trajectories and integrate it into the world model, we can leverage it to provide accurate predictions of the robot dynamics when learning new tasks, without requiring significant training data.

However, the intricate interactions between the robot and objects in the environment make it challenging to learn intrinsic robot dynamics in isolation. For example, when the robot operates the lever switch, it encounters resistance from the lever, and when it pushes all the way down, it can no longer move forward. Additionally, the lever itself moves as a result of the robot's movement. This interplay complicates learning generalized robot dynamics that are applicable across various tasks.

Therefore, we propose a simplified interaction framework as shown in Figure 2. We decouple world dynamics into robot dynamics and environment dynamics. The robot dynamics need to take into account the intrinsic dynamics of robot itself as well as potential effects of the environment on the robot, such as resistance to encountering an object, with no need to know the detailed environmental state transitions. The environment dynamics then infer the movement of objects in the environment based on the execution results of the robot rather than directly through action signals. This framework simplifies the robot-object interaction by explicitly modeling the robot's effects on the objects while implicitly modeling the objects' influences on the robot.

Based on this framework, we allow inference on robot dynamics to be relatively independent of the environment dynamics, thus learning intrinsic robot dynamics that can be generalized across a variety of tasks. For example, as shown in Figure 2(a), the robot first executes the action according to the control signal, and it does not need to know how the lever moves, but only needs to know that the reaction force of the lever will be applied when it pushes forward. We denote the potential effects on the robot by the lever switch at the edge of the dashed line. Then, based on the robot's execution results, we can infer the specific movement track of the lever switch. Eventually, we combine the two inference results to obtain the complete state estimation.

### 4.2 ROBO-CENTRIC WORLD MODELS

Based on the above idea, we introduce a novel architecture called Robo-Centric World Model (RCWM), which comprises two branches to learn robot dynamics and environment dynamics separately, and an interaction model leveraging the cross-attention mechanism to assess the effect of robot dynamics predictions on the environment state. RCWM builds upon DreamerV3 and an overview of the architecture is shown in Figure 3. We describe the components of it in detail below.

**Representation model** To ensure that each branch captures relevant features, we first utilize a shared encoder to learn low-level features. Subsequently, distinct encoders are constructed for each branch to capture their respective features. We denote the robot branch with the superscript $R$ and the environment branch with the superscript $E$. The stochastic representations are calculated as follows:

$$z_t^R \sim q_\varphi(z_t^R \mid h_t^R, o_t) \quad z_t^E \sim q_\phi(z_t^E \mid h_t^E, o_t) \tag{4}$$

**Dynamics model** Based on the simplified interaction framework described above, we design the robot dynamics model to be action-conditioned, which updates the next recurrent state $h_t^R$ based on the latent state $[h_{t-1}^R, z_{t-1}^R]$ and the action $a_{t-1}$. The environment dynamics model does not directly take actions as inputs; instead, it uses the interaction features $e_{t-1}$, which are calculated by the interaction model. Then we can get the stochastic representation based on the recurrent state:

$$h_t^R \sim f_\varphi(h_{t-1}^R, z_{t-1}^R, a_{t-1}) \quad h_t^E \sim f_\phi(h_{t-1}^E, e_{t-1}) \tag{5}$$

$$\hat{z}_t^R \sim p_\varphi(\hat{z}_t^R \mid h_t^R) \quad \hat{z}_t^E \sim p_\phi(\hat{z}_t^E \mid h_t^E) \tag{6}$$

**Interaction model** The interaction model aims to calculate how the robot's movements will affect the environment. This design allows the environment dynamics to be predicted not directly with actions, but with the effect of the robot's movements results on the environment, thus allowing the environment dynamics to derive state predictions that are consistent with the robot's movements. Specifically, after the robot branch predicts the next latent state $\hat{s}_t^R$ according to the action, we seek to capture the effect this movement will induce in the current representation of the environment $z_{t-1}^E$. To learn about this interaction, we construct a model with the cross-attention mechanism, generating keys and values from $\hat{s}_t^R$ and queries from $z_{t-1}^E$. We then add the attention result $x$ and $z_{t-1}^E$ to get the interaction feature $e_{t-1}$. See Appendix A.2 for implementation details. The interaction model is

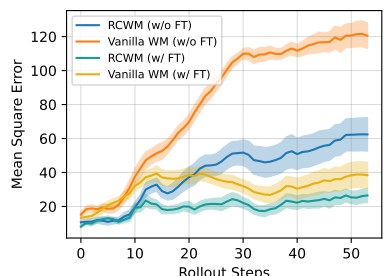

(a) Robot dynamics prediction error visualization    (b) Robot dynamics prediction error curves

Figure 4: (a) Robot dynamics prediction error of pre-trained RCWM on the unseen *coffee-pull* task. (b) Robot dynamics prediction error curves for each step of the imaginary trajectory. The solid line and shaded regions represent the mean and standard deviation, respectively, across 100 imaginary trajectories. See Appendix A.5.2 for full curves for all tasks.

calculated as follows:

$$Q = g_\phi(z_{t-1}^E) \quad K = V = g_\varphi(\hat{s}_t^R) \quad x = Attention(Q, K, V)$$
$$e_{t-1} = LayerNorm(x + z_{t-1}^E) \tag{7}$$

**Mask-guided decoder**    To enable the two branches to capture their respective relevant information from high-dimensional image observations, we design a mask-guided reconstruction mechanism to introduce structured prior robot masks inspired by Gmelin et al. (Gmelin et al., 2023). We construct independent image decoders for each branch, which take latent state as input and output the reconstructed image with a sigmoid-activated mask. Then we multiply the reconstructed image with the mask and combine the results from both branches to produce the final reconstructed observation:

$$\hat{o}_t^R, \hat{m}_t^R \sim p_\varphi(\hat{o}_t^R, \hat{m}_t^R \mid s_t^R) \quad \hat{o}_t^E, \hat{m}_t^E \sim p_\phi(\hat{o}_t^E, \hat{m}_t^E \mid s_t^E)$$
$$\hat{o}_t = \hat{o}_t^R * \hat{m}_t^R + \hat{o}_t^E * \hat{m}_t^E \tag{8}$$

Then we construct an auxiliary loss that guides the robot branch to capture relevant features by minimizing the mean squared error between the prediction robot mask $\hat{m}_t^R$ and the prior robot mask $m_t^R$. With the assistance of prior robot masks, we implicitly achieve feature disentanglement through the decoder, rather than explicitly processing the image input. This allows the robot branch to focus on robot-related features without entirely disregarding environmental information, seamlessly aligning with our simplified interaction framework, in which robot dynamics need to implicitly take into account the potential effects of the environment. This mechanism allows the world model to naturally utilize the prior robot mask to provide additional structured information for the learning of robot dynamics. Moreover, by using a few prior robot masks to guide the warm-up of RCWM, it can provide stable and accurate mask predictions for subsequent training without relying on prior masks. In addition, we find that this architecture naturally captures structured information even without the prior mask, which we provide more details in Appendix A.5.1.

**Predictor Heads**    The latent state of both branches contains only partial information relevant to their respective purpose. To provide comprehensive information to the reward predictor and the continue predictor, we combine them to form the final latent state $s_t \doteq [s_t^R, s_t^E]$. This combined latent state is also utilized for behavior learning.

Overall, RCWM can be jointly optimized by the following loss:

$$\mathcal{L}^{RC} \doteq \mathbb{E}\Big[\sum_{t=1}^{T} \Big( -\ln p(o_t \mid s_t^R, s_t^E) - \ln p(r_t \mid s_t) - \ln p(c_t \mid s_t) + (\hat{m}_t^R - m_t^R)^2$$
$$+ \mathcal{L}_{KL}(q_\varphi(z_t^R \mid h_t^R, o_t), p_\varphi(\hat{z}_t^R \mid h_t^R)) + \mathcal{L}_{KL}\big(q_\phi(z_t^E \mid h_t^E, o_t), p_\phi(\hat{z}_t^E \mid h_t^E)\big)\Big)\Big]. \tag{9}$$

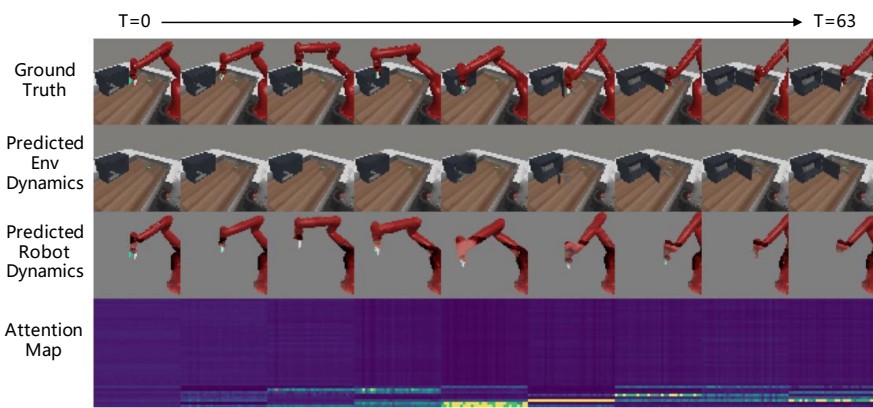

Figure 5: Interaction predictions of pre-trained RCWM on *door-open* task. The complete imaginary trajectory with the length of 64 steps is shown from left to right at 8-step interval.

### 4.3 IMPLEMENTATION PIPELINE

In this subsection, we introduce the pre-training and fine-tuning pipeline for our proposed Robo-Centric World Model, as shown in Figure 1. Note that unlike previous approaches, we concentrate on pre-training on upstream tasks that are entirely different from downstream tasks but involve the same robot.

First, we need to obtain a few prior robot masks to guide the warm-up of RCWM. We randomly select a small number of trajectories from the pre-training dataset formed by the trajectories collected from upstream tasks. In our experiments, we sample 16 full trajectories totaling 4000 steps. Then we segment the robot in the image to obtain the mask with the help of SAM2. SAM2 can automatically track segmented objects, so we only need to provide detection points for the first frame of the trajectory. Since the initial position of the robot is fixed, we only need to set identical positive and negative points for the first frame of all trajectories. We provide details in Appendix A.4. Then we utilize these trajectories with masks to warm up the RCWM, thus ensuring that both the robot branch and the environment branch capture the corresponding features. Since RCWM is naturally capable of separating structured information, it does not heavily rely on prior masks, thereby avoiding excessive costs. In cases where prior masks are readily available, e.g., directly from the environment simulator, we can use them throughout the pre-training phase to guide training of RCWM, thus enhancing robot dynamics learning.

After warmup, RCWM already has the ability to capture robot features. With mask reconstruction mechanism, RCWM is still able to accurately predict robot masks without using prior masks. Therefore, we can automatically learn robot dynamics by mask free pre-training of RCWM on the full pre-training dataset. After pre-training, the robot branch has been able to accurately predict the robot's movements based on actions. We keep the robot dynamics model and reset the environment dynamics and interaction models to prepare for learning new tasks. The robot branch is not affected by the reset of these two models, as its inference is completely independent of the environment branch.

During the online fine-tuning phase, although the environmental observations change in new tasks, the robot branch remains capable of robustly predicting the robot mask, allowing for the fine-tuning of RCWM without using prior robot masks. Despite being well-trained, the robot branch still needs to learn about the potential effects of the environment on the robot when faced with new tasks. With online fine-tuning of the RCWM overall, it is able to quickly adapt to the dynamics of new tasks and learn about the robot-object interaction.

## 5 EXPERIMENTS

In this section, we aim to explore the advantages of RCWM and answer the following questions: (1) Can RCWM effectively learn generalized robot dynamics applicable to various new tasks? (2) Can

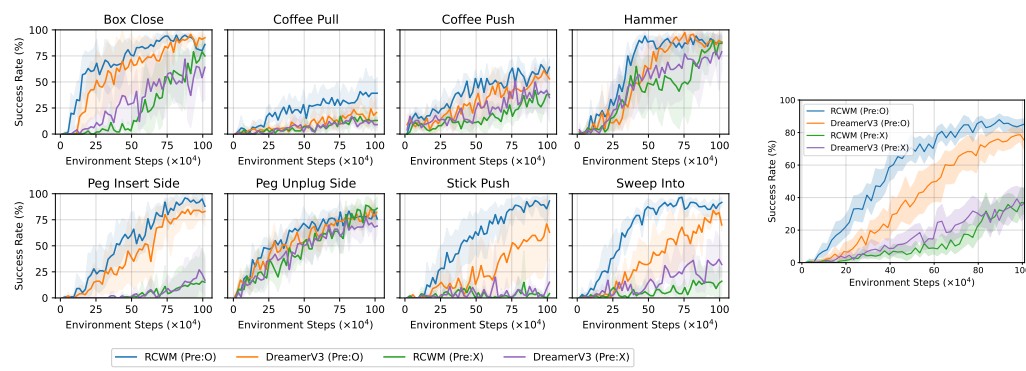

(a) Learning curves of fine-tuning on 8 Meta-world tasks      (b) Aggregate performance

Figure 6: Fine-tuning results on Meta-world. (a) The solid line and shaded regions represent the mean and bootstrap confidence intervals, respectively, across five runs. (Pre: O) represents pre-training usage and (Pre: X) indicates no pre-training. (b) Aggregate performance across a total of 40 runs over 8 tasks.

RCWM accurately learn the interactions between the robot and objects? (3) Can pre-trained RCWM improve sample efficiency for learning new tasks? (4) Can RCWM provide robust representations for the policy in the presence of environmental visual disturbances?

## 5.1 EXPERIMENTAL SETUP

**Environments**  Our approach is focused on visual robot control scenarios that utilize the same robot for various manipulation tasks. Therefore, we select the Meta-world benchmark (Yu et al., 2020) to evaluate our approach, which includes 50 manipulation tasks performed with the Sawyer robot and is friendly to reinforcement learning. We first select 4 simple tasks for pre-training data collection. Then we select 8 challenging tasks for sample efficiency evaluation. All of these tasks involve complex robot-object interactions and require precise manipulation, which places significant demands on the predictive accuracy of world models. we set the action repeat to 2 and the episode length to 500 for all tasks. See Appendix A.3 for detailed task description and parameter setting.

**Pre-training dataset**  We utilize the replay buffer from 4 tasks trained by DreamerV3 as our dataset, with each replay buffer containing 500K steps of interaction data. We randomly selected 16 full trajectories from the pre-training dataset and then used SAM 2 to generate a total of 4k masks for mask-guided warmup.

**Baseline**  We choose Dreamerv3 with vanilla world models (WM) as our baseline since our approach is built upon it. As RCWM utilizes prior robot masks, we also add a mask predictor to the vanilla world model for fairness. The pre-training process for vanilla WM is identical to that of RCWM, except that it does not require resetting the dynamic model after pre-training.

**Implementation details**  Since Meta-world features hard-exploration tasks, this leads to significant variance in the learning curves, making the evaluation of sample efficiency challenging. In addition, exploring randomness can seriously impact the assessment of pre-trained world models. This is because trying to explore a successful trajectory often requires numerous interactions with the environment, causing the advantages of pre-training to diminish as online interactions and training progress. To mitigate these problems, we randomly select 20 successful trajectories to pre-fill the replay buffer at the beginning of training. This practice is effective in improving the stability of training on Meta-world tasks (Wu et al., 2024).

## 5.2 EXPERIMENTAL RESULTS

**Robot dynamics learning**  We evaluate the effectiveness of RCWM in learning robot dynamics based on the prediction accuracy. We collect 100 trajectories containing rich robot dynamics for

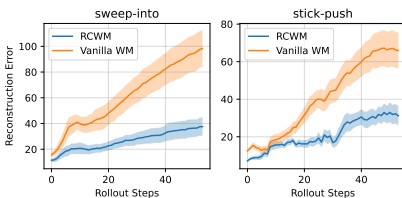 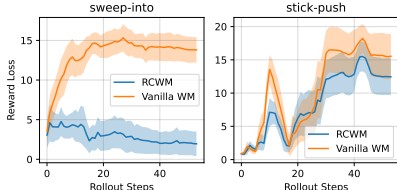

(a) Comparison of reconstruction errors.  (b) Comparison of reward loss.

Figure 7: Comparison of the reconstruction error and the reward loss for imaginary trajectories generated on the evaluation dataset. The solid line and shaded regions represent the mean and standard deviation, respectively, across 100 imaginary trajectories.

each task using well-trained policies as the evaluation dataset and use the world model to generate imaginary trajectories for comparison. We multiply the imagined reconstructed image with the ground-truth robot mask and compute the mean square error with the ground-truth robot image to serve as the robot dynamics prediction error. As shown in Figure 4, even without fine-tuning, the pre-trained RCWM is still able to consistently delivers more accurate predictions of robot movements in response to action signals compared to the vanilla WM when applied to unseen tasks. After fine-tuning, the vanilla WM still does not achieve the same level of accuracy in predicting robot dynamics as RCWM. See Appendix A.5.2 for complete results evaluated on all tasks. The results indicate that explicitly decoupling and learning robot dynamics individually can lead to more accurate dynamics predictions than using a single dynamics model to learn the whole world dynamics.

**Robot-object interaction learning**   We analyze the learning of the interaction between robot and objects according to the consistency of movement between the predictions of the environment branch and the robot branch. We select the *door-open* task to demonstrate this, as it involves significant interactions. Figure 5 illustrates the prediction of object movements in the environment as the robot moves. We can see that RCWM successfully learns the interaction between the robot and the door. We also visualize the attention map in the interaction model, with detailed descriptions and additional visualizations provided in Appendix A.5.3. Our observations indicate that the interaction model does not simply feed all information from the robot branch into the environment branch, as the attention map is only significantly activated at certain locations. In addition, we notice an interesting phenomenon where the attention map seems to be significantly activated when the robot induces changes in the environment, such as when it first touches or grabs the object. This suggests that the interaction model does learn certain physical interaction relationships. We provide visualizations for other tasks in Appendix A.5.3. The results demonstrate the effectiveness of our proposed simplified interaction framework and RCWM, which not only decouples robot dynamics but also models physical interactions with objects.

**Sample efficiency improvement**   Figure 6 shows the learning curves comparing RCWM and vanilla WM across 8 challenging tasks in Meta-world. The results indicate that pre-trained RCWM can significantly improve sample efficiency compared to DreamerV3 with vanilla world model. We consider that this is due to the accurate prediction of the robot dynamics provided by the pre-trained RCWM, which allows it to quickly adapt to new tasks and generate accurate imaginary trajectories. Although the prediction of environment dynamics is also crucial for reward estimation, only accurate prediction of the robot's movements can better predict the interaction with the environment. Since the actor and critic are based on imaginary trajectories for policy improvement and value estimation, the accuracy of the generated trajectories is crucial, especially for tasks that require fine manipulation. We evaluate the reconstruction errors and the reward loss of imaginary trajectories, and the results are presented in Figure 7. There is no significant final performance gap between the policies learned respectively on the two world models used for the comparison, ensuring fairness. See AppendixA.5.4 for comparisons of other tasks. We can see that RCWM generates more accurate imaginary trajectories and reward predictions than vanilla WM on almost all tasks, especially on *sweep-into* and *stick-push*, both of which require fine control of the robot. This explains the improvement in sample efficiency achieved by our approach.

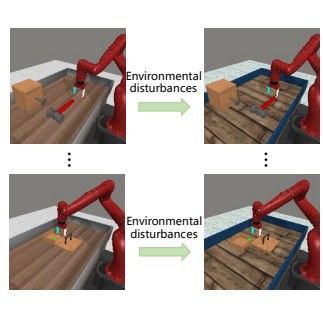

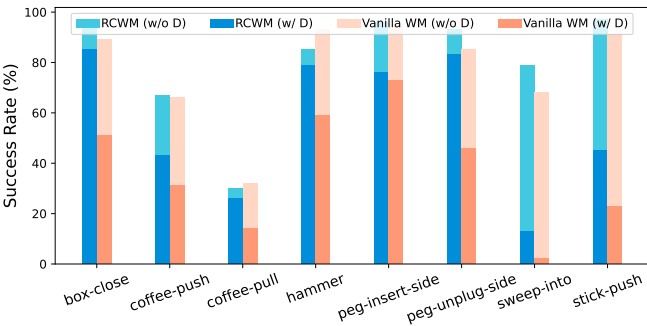

(a) Visualization of environmental disturbances.

(b) Performance comparison with disturbances

Figure 8: Robustness evaluation with environmental visual disturbances. Each task is evaluated 100 times. The *D* in the legend of Figure 8(b) denotes disturbances.

**Robustness against environmental disturbances**    Considering that one feature of RCWM is the ability to learn generalized robot dynamics individually without paying too much attention to environment information, we would like to evaluate whether it can still provide robust representation and dynamics prediction of the robot against environmental visual disturbances. This is crucial for the robustness of the policy. We change the texture of some objects in the environment to introduce disturbances, as shown in Figure 8. We visualize the results of the fine-tuned RCWM in predicting robot dynamics when faced with unseen texture changes, see Appendix A.5.5. We find that the robot branch in RCWM is not sensitive to environmental disturbances and is still able to accurately predict the robot's movements, even when replacing the background with random noise. To illustrate whether RCWM can provide robust representations for the policy, we select policies with similar performance that are trained with RCWM and vanilla WM, respectively, to evaluate the change in performance after the disturbance. As shown in Figure 8(b), the policy trained with vanilla WM suffers severe performance drop when faced environmental disturbances, while the policy trained with the RCWM exhibits robustness on almost all tasks. We propose that this is because our policy learning concatenates the respective representations of the robot and the environment as inputs. Even if the environmental representation is disturbed to some extent, the policy can still make decisions based on the robust robot representation, thereby enhancing its overall robustness.

## 6    CONCLUSION AND DISCUSSION

In this paper, we introduce the Robo-Centric World Model (RCWM), which can decouple the dynamics of the robot and the environment, and learn their interaction via an interaction model based on the cross-attention mechanism. With RCWM, we can extract robot dynamics through pre-training on upstream tasks with the assistance of a few prior robot masks, and use this prior knowledge about robot dynamics to improve sample efficiency on downstream tasks. Compared to vanilla world models, the RCWM has significant advantages in that it can be efficiently pre-trained to learn generalized robot dynamics applicable to various new tasks and can remain robust against noise disturbances. In addition, we provide an efficient way to introduce external prior robot masks into the training of the world model to enhance robot dynamics learning. Our experiments on Meta-world support our argument that integrating prior knowledge of robot dynamics into world models can effectively improve sample efficiency for downstream visual robot control tasks.

Despite these results, some limitations still remain. Our approach focuses on visual robot control tasks and is difficult to effectively extend to more general tasks such as video games or autonomous driving. In addition, our approach requires using the same robot's interaction data for pre-training and cannot be directly expanded to general in-the-wild robot data. However, we still believe that we provides a novel idea for constructing world models applicable to robot manipulation tasks, which can further improve the sample efficiency for MBRL. In future work we would like to explore utilizing the long horizon predictive accuracy of the RCWM for robot manipulation planning.

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

# A APPENDIX

## A.1 BEHAVIOR LEARNING DETAILS

We follow the actor-critic learning scheme of DreamerV3 as described below:

$$\text{Actor:} \quad a_t \sim \pi_\theta(a_t \mid s_t) \qquad \text{Critic:} \quad v_\psi(s_t) \approx \mathbb{E}_{p_\varphi, p_\phi, \pi_\theta}[R_t]$$
$$R_t^\lambda \doteq r_t + \gamma c_t \left((1-\lambda)v_\psi(s_{t+1}) + \lambda R_{t+1}^\lambda\right) \quad R_T^\lambda \doteq v_\psi(s_T) \tag{10}$$

The world model generates imaginary trajectories $\{s_{1:T}, a_{1:T}, r_{1:T}, c_{1:T}\}$ starting from the sampled latent state with the actor. To estimate returns that consider rewards beyond the prediction horizon, bootstrapped $\lambda$-returns (Sutton & Barto, 2018; Schulman et al., 2015) are calculated to integrate the predicted rewards and the values. The critic learns to predict the distribution of the return estimates $\lambda$-target $R_t^\lambda$ using the maximum likelihood loss:

$$\mathcal{L}(\psi) \doteq -\sum_{t=1}^{T} \ln p_\psi\left(R_t^\lambda \mid s_t\right) \tag{11}$$

The actor learns to select actions that maximize the return while exploring through an entropy regularizer. Since we deal with a continuous action space, we use stochastic backpropagation following DreamerV3 to estimate the gradient of the following loss:

$$\mathcal{L}(\theta) \doteq \sum_{t=1}^{T} \mathbb{E}_{\pi_\theta, p_\phi}\left[-\operatorname{sg}\left(R_t^\lambda\right) / \max(1, S)\right] - \eta \mathrm{H}\left[\pi_\theta\left(a_t \mid s_t\right)\right]$$
$$S = \operatorname{Per}\left(R_t^\lambda, 95\right) - \operatorname{Per}\left(R_t^\lambda, 5\right) \tag{12}$$

where sg is a stop gradient function and $S$ represents the return scale which is calculated using the average of the exponential decay from the 5th batch percentile to the 95th batch percentile. We set entropy scale $\eta = 3 \cdot 10^{-4}$ in all experiments. For more details about DreamerV3 we refer to Hafner et al. (2023)

## A.2 INTERACTION MODEL DETAILS

The stochastic representation $z_t \in \mathbb{R}^{n \times m}$ is a vector of multiple categorical variables following DreamerV3, where $n$ is the number of categoricals and $m$ is the dimension of each categorical variable. In our experiments we set $n = m = 32$. In order to fully utilize the predictive information about the robot dynamics, we take $\hat{s}_t^R = \{h_t, \hat{z}_t^R\}$ as input to compute its relationship with $z_{t-1}^E$. Since the dimension of $h_t \in \mathbb{R}^d$ is different from $\hat{z}_t^R$, we first input $h_t$ into a MLP and reshape the output to get $g_t \in \mathbb{R}^{k \times m}$, where in our experiment $k = 8$. Then we concatenate the two together to get $u_t^R \in \mathbb{R}^{(n+k) \times m}$. We generate queries using $z_{t-1}^E$ and keys and values using $u_t^R$. We construct the cross-attention model with output dimension $m$ to get $x_t \in \mathbb{R}^{n \times m}$. To ensure the completeness of the environment information, we add $z_{t-1}^E$ and $x_t$ to get the final interaction feature $e_{t-1} \in \mathbb{R}^{n \times m}$.

## A.3 TASK SETTINGS

We report the specific tasks in our experiment in Table 1 and show all the task scenarios in Figure X. All tasks use the v2 version and the "goal-observable" mode [1]. We refer to Seo et al. (Seo et al., 2023) to adjust the camera position to $[0.75, 0.075, 0.7]$. We set the maximum length of the episode to 500 environment steps with the action repeat set to 2. The fine-tuning task is incredibly difficult compared to the pre-training task. This setup can demonstrate the characteristics of our approach, which is able to extract task-irrelevant robot dynamics to accelerate training of various downstream tasks.

---

[1]https://github.com/Farama-Foundation/Metaworld

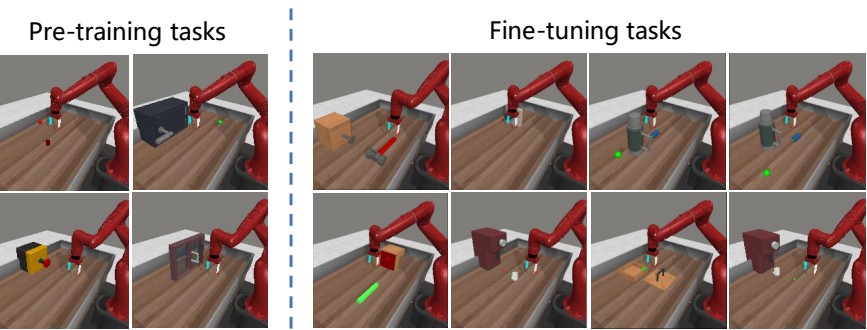

Figure 9: Visualization of task scenarios.

Table 1: Specific tasks in Meta-world used in our experiments.

|  | Task name | |
| --- | --- | --- |
| Pre-training tasks | button-press
door-open | window-open
reach |
| Fine-tuning tasks | box-close
coffee-pull
peg-insert-side
sweep-into | coffee-push
hammer
peg-unplug-side
stick-push |

## A.4 ROBOT MASK GENERATION

We utilize Segment-Anything-Model 2 (SAM 2) to generate robot masks. Since SAM 2 has the ability to track and segment objects in the video, we select complete trajectories rather than fragments, thus only needing to give positive and negative points to the first frame of each trajectories. We find that there is no randomization of the initial position of the robot in Meta-world, so the initial position of the robot is fixed for each trajectory. In our selected tasks, there are a total of two initial positions involved. We manually create two sets of prompt points, as shown in Figure 10. The positive points for the robot cover the robot's body as well as the gripper, while the negative points are used to avoid segmentation failures due to the object's initial position being too close to the gripper. In order for SAM 2 to generate masks accurately, we additionally save high-quality image observations obtained from the simulator with a resolution of 256*256. After generated the mask, we resize it to 64*64 for model training. This can effectively improve the accuracy of mask generation. In our tests, SAM 2 performs well in all tasks.

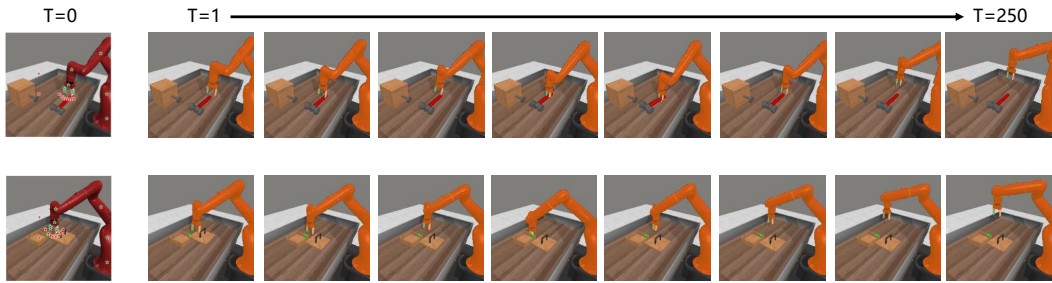

Figure 10: Visualization of robot mask generation with SAM 2. The green markers in the first frame indicate positive points and the red markers indicate negative points.

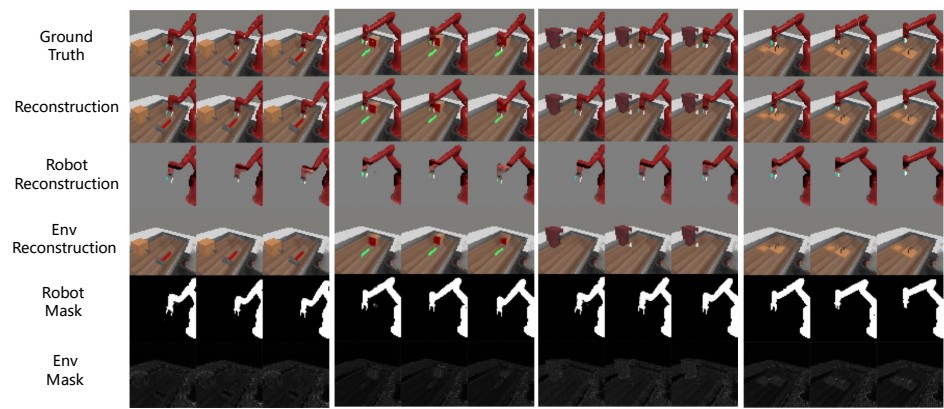

Figure 11: Visualization of RCWM reconstruction results.

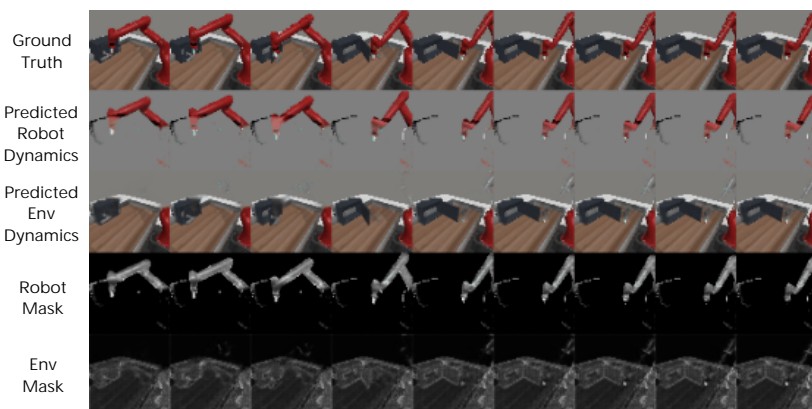

Figure 12: Reconstruction results on the *door-open* task that training without prior robot masks.

## A.5 SUPPLEMENTAL EXPERIMENT RESULTS

### A.5.1 MASK-GUIDED REPRESENTATION DISENTANGLEMENT

With the help of prior robot masks, RCWM can disentangle robot-related information and environment-related information from high-dimensional image inputs. Figure 11 shows the reconstruction results of two branches of RCWM on several tasks. In our experiments, we also notice that without the auxiliary loss about the prior mask, both branches of RCWM still capture some structured information. We show the results of the reconstruction of RCWM in Figure 12, which is not trained with the prior mask. We can see that each of the two branches captures different structured information. However, without the prior information, we find that we cannot control with certainty exactly what information is captured by the two branches. Sometimes robot-related information is captured by the environment branch. Despite the problems, this implies that the architecture of the RCWM has the potential for representation disentanglement. This problem is solved by using the auxiliary task with the prior robot mask for warm-up.

### A.5.2 ROBOT DYNAMICS LEARNING

We complement the visualization of the robot dynamics prediction error using the pre-trained RCWM on the unseen hammer task as shown in Figure 14. Figure 13 illustrates the curve of prediction error of robot dynamics for all tasks on the evaluation dataset. The evaluation dataset is collected using the well-performing policies that have been fine-tuned on each tasks, and robot masks have been generated for all data using SAM2. During the evaluation of errors, we select the trajectories corresponding to each task, using the first 10 steps of data as historical observations, and

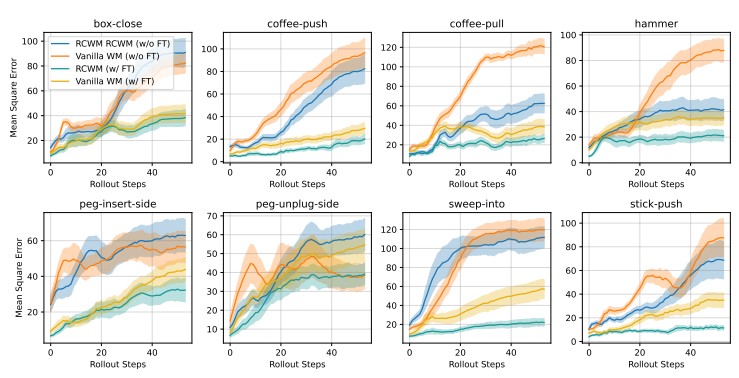

Figure 13: Robot dynamics prediction error curves on all tasks.

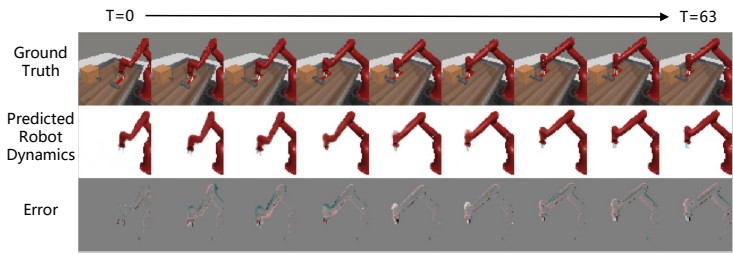

Figure 14: Visualization of robot dynamics prediction error.

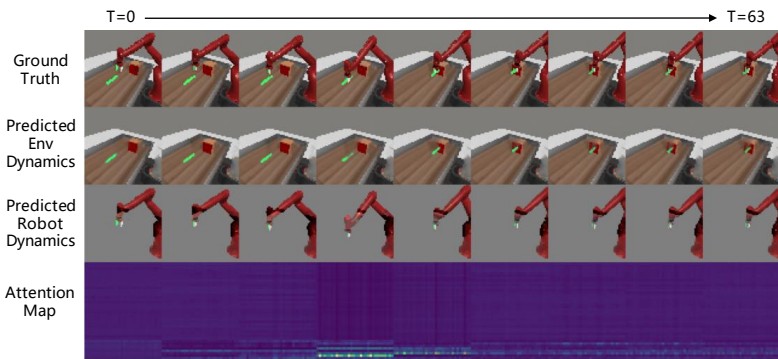

Figure 15: Interaction prediction results on Meta-world *peg-insert-side* task.

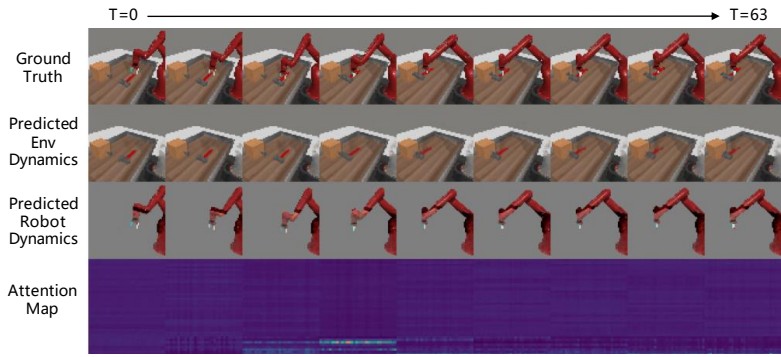

Figure 16: Interaction prediction results on Meta-world *hammer* task.

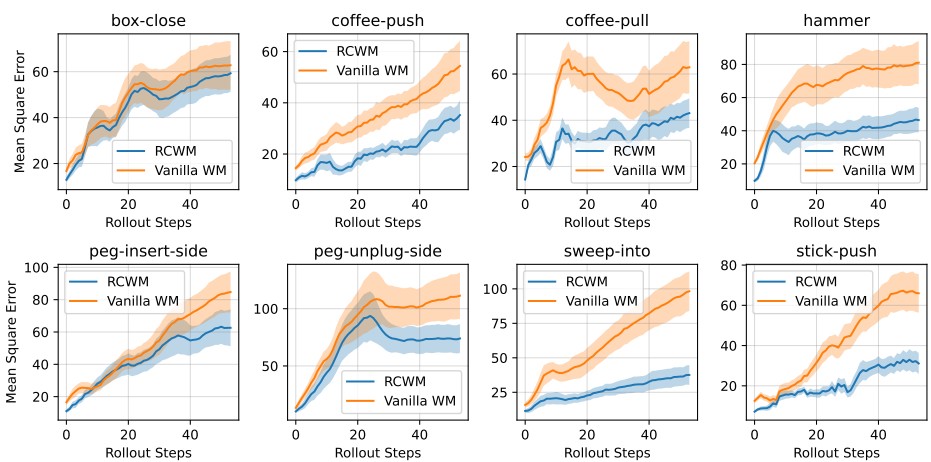

Figure 17: Comparison of reconstruction errors.

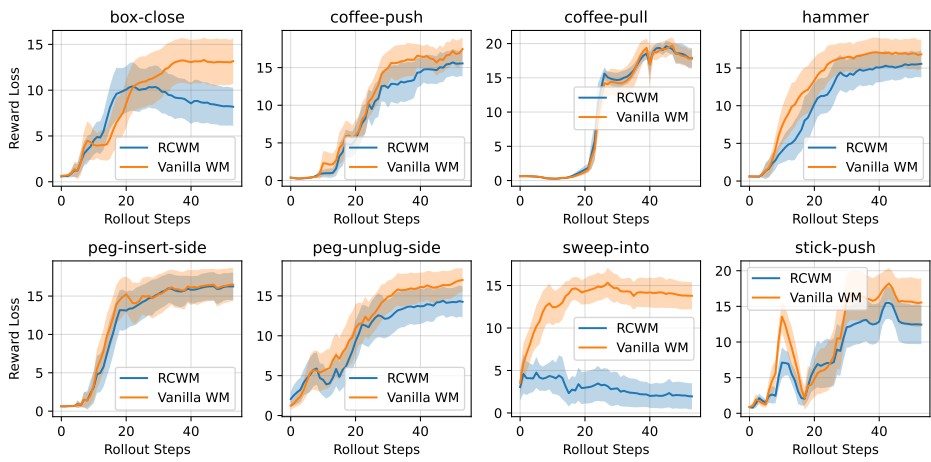

Figure 18: Comparison of reward loss.

then utilize the world model to predict the next 54 steps of states based on the actions. We calculate the error for all trajectories and compute the mean and standard deviation.

### A.5.3 ROBOT-OBJECT INTERACTION LEARNING

We supplement some interaction prediction results from fine-tuned RCWM as shown in Figure 15 and Figure 16. It can be seen that RCWM is able to capture the interaction between robots and objects and can make accurate predictions. In addition, we visualize the attention map in the inter-action model to show its activation in different states. The shape of the attention map is $40 \times 32$, where the $32 \times 32$ map at the top represents the attention between $z_{t-1}^E$ and $\hat{z}_t^R$, and the $8 \times 32$ map at the bottom represents the attention between $z_{t-1}^E$ and $\hat{h}_t^R$. We reshape it to $80 \times 64$ in the figure to align the size of the observation. It can be seen that the interaction model mainly captures the information in $\hat{h}_t^R$, which contains information about the robot's history as well as the current action. Since $\hat{z}_t^R$ is a stochastic representation computed from $\hat{h}_t^R$, when the environment is not stochastic, the information contained in $\hat{z}_t^R$ is not as comprehensive as $\hat{h}_t^R$. However, given the consistency of the information, we still compute attention using $\hat{s}_t^R = \{\hat{z}_t^R, \hat{h}_t^R\}$.

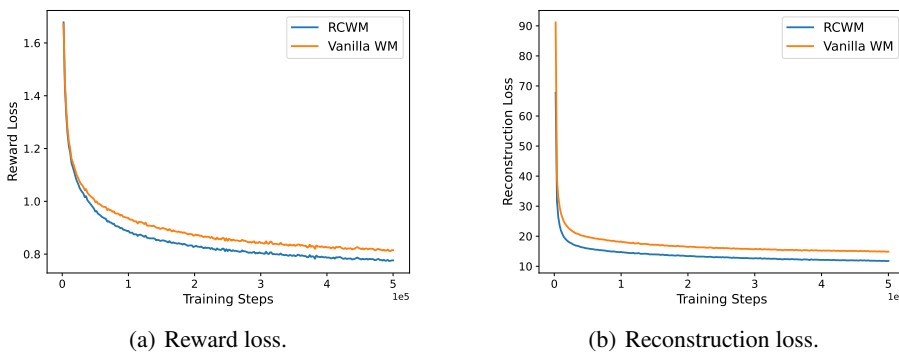

(a) Reward loss.  (b) Reconstruction loss.

Figure 19: Comparison of the reward loss and reconstruction loss during pre-training.

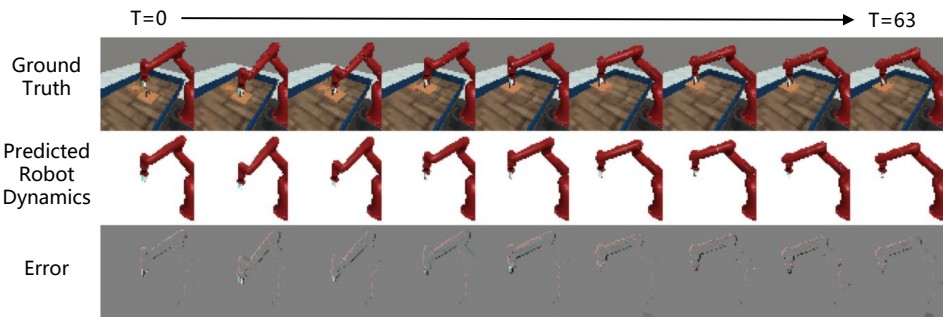

Figure 20: Robot dynamics prediction error visualization with environmental texture changes.

#### A.5.4 SAMPLE EFFICIENCY IMPROVEMENT

We supplement the comparison of the reconstruction error and the reward loss for imaginary trajectories in all tasks, as shown in Figure 17 and Figure 18. We select models that perform well after fine-tuning and have no significant performance gaps for evaluation. It can be seen that RCWM is able to generate more accurate trajectories and significantly outperforms the vanilla world model in long horizon prediction. We will try to utilize this long-horizon prediction capability for planning in our future work. In addition, we find that RCWM has lower reconstruction loss and reward estimation loss during pre-training, as shown in Figure 19.

#### A.5.5 ROBUSTNESS AGAINST ENVIRONMENTAL DISTURBANCES

We visualize the predicted robot dynamics of the fine-tuned RCWM when facing environmental disturbances, as shown in Figure 20 and Figure 21. We find that it is still effective in predicting the robot's movement based on action.

### A.6 CODE IMPLEMENTATION

Our code references the official DreamerV3 code [2] and the PyTorch implementation [3]. Our code is available in the anonymous repository: https://github.com/robo-centric-wm/robo-centric-world-model.

---

[2] https://github.com/danijar/dreamerv3
[3] https://github.com/NM512/DreamerV3-torch

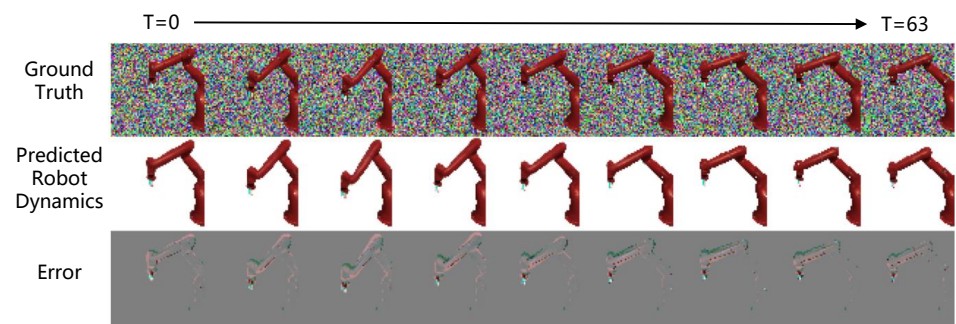

Figure 21: Robot dynamics prediction error visualization with random noise disturbances.

## A.7 COMPUTATIONAL RESOURCES

We use a single Nvidia RTX3090 GPU and 10 CPU cores for each training run. In terms of parameter counts, RCWM consists of 24.7M parameters, while the vanilla world model in DreamerV3 consists of 19.1M parameters. In terms of training time, it takes ~50 hours for pre-training of RCWM over 500K updates and ~90 hours for fine-tuning of RCWM with MBRL for each run of Meta-world experiments over 1M environment steps. This takes longer than training of vanilla DreamerV3, which requires ~60 hours for pre-training and ~80 hours for fine-tuning. We consider that this is mainly because RCWM creates two RNNs that compute in sequential order. In terms of memory usage, RCWM requires ~5GB GPU memory for each run of Meta-world experiments, while vanilla DreamerV3 requires ~4GB GPU memory.

## A.8 HYPERPARAMETERS

We report the hyperparameters used in our experiments in Table 2. Unless otherwise specified, we use the same hyperparameters as DreamerV3 (Hafner et al., 2023). In the pre-training phase, we set the number of pre-training update steps to 500k, at which point the model has converged. To avoid performance gains due to the increase in the number of model parameters, we reduce the hidden dimension of the dynamic model in RCWM to 256, while the vanilla world model is 512. In order to improve training stability, we increase the scale of the reward loss to 100 with reference to Ma et al. (2024). We set the mask-guided warmup steps to 1000, which is much larger than DreamerV3's original pre-training step of 100 after random prefilling. This is because we want to make RCWM learn to capture robot-related features with only a few prior robot mask data.

Table 2: Hyperparameters settings in our experiments. Unless otherwise specified, we use the same hyperparameters used in DreamerV3 (Hafner et al., 2023)

| | Hyperparameter | Value |
|---|---|---|
| | Image observation | $64 \times 64 \times 3$ |
| | Action repeat | 2 |
| | Max episode length | 500 (Meta-world) |
| | Evaluation episodes | 20 |
| | Random exploration | 2500 steps |
| | Replay buffer capacity | $1 \times 10^6$ |
| | Batch size | 16 |
| | Batch length ($T$) | 64 |
| Base settings | Imagination horizon ($L$) | 15 |
| | Discount ($\gamma$) | 0.997 |
| | Return lambda ($\lambda$) | 0.95 |
| | World model learning rate | $1 \times 10^{-4}$ |
| | Actor-critic learning rate | $3 \times 10^{-5}$ |
| | Actor entropy scale ($\eta$) | $3 \times 10^{-4}$ |
| | Dynamics hidden dimension | 256 |
| | Reward loss scale | 100 |
| | Pre-training steps | $5 \times 10^5$ |
| Pre-training | Replay buffer capacity | $2 \times 10^6$ |
| | Mask-guided warmup steps | 1000 |
| | Generated mask number | 4000 |
| | Expert trajectory number | 20 |
| Fine-tuning | Mask-free warmup steps | 500 |
| | Fine-tuning steps | $1 \times 10^6$ |

