# OpenReview forum: "Pre-Training Robo-Centric World Models For Efficient Visual Control"
_ICLR.cc/2025/Conference — Submitted to ICLR 2025_

### Official Review · Reviewer_tbDh · 2024-10-29

**Soundness:** 3
**Presentation:** 3
**Contribution:** 2
**Rating:** 8
**Confidence:** 4

**Summary:**

This paper introduces the Robo-Centric World Model (RCWM), which decouples robot dynamics from environmental dynamics, and employs an interaction model to evaluate the impact of the robot's actions on the environment. The authors use SAM2 to create segment masks for the robot, which are then used to warmup the mask reconstruction process. The subsequent pre-training is conducted without mask usage. Experiments are conducted in the Meta-world domain, demonstrating through quantitative and visual results that RCWM can learn distinct dynamics and show robustness against disturbances.

**Strengths:**

- The decomposition of robot and environmental dynamics is a straightforward method that enables transfer of shared knowledge.
- The distillation of SAM into world models that enables the agent to distinguish robot and background is an interesting practice.
- The experimental results show improvement over the base model, with the rich visualization results offering valuable insights.

**Weaknesses:**

- Discussion and comparison to related works are missing. The authors state that action-free pre-training methods "are of limited help when confronted with robot manipulation tasks that require accurate predictions". However, the paper only carries out experiments in the Meta-world domain, where other action-free pre-training methods also demonstrate good performance.
- Limitations are not addressed adequately. RCWM limits the interaction between the robot and the environment from two-way to one-way, which is acceptable for Meta-world since pre-training and fine-tuning are carried out on the same platform. However, RCWM may encounter difficulties when applied to certain types of real-world visual robotics tasks, such as dealing with a different friction coefficient or a different tilting angle for the table.

**Questions:**

- In Figure 7(a), do RCWM and vanilla WM use the same set of trajectories for evaluation, or are the trajectories sampled independently for each model? If the models share the same trajectories, how are they sampled?
- In Figure 7(b), a sharp turn can be observed between 10 and 20 rollout steps for the stick-push task. Why would this happen?
- A discussion section comparing RCWM to previous action-free pre-training methods should be added. The authors should compare RCWM to previous action-free pre-training methods specifically on robot manipulation tasks that requires accurate predictions.
- A limitation section about the applicability and generality scope of RCWM should be added. The authors should explicitly discuss the potential limitations in applying RCWM to real-world scenarios with varying physical properties.

Minor comments:
- The notation for the baseline method is a bit inconsistent. I suggest changing all "vanilla WM" notations to "DreamerV3" or vice versa for better readability.
- Typo: "dynamic model" should be "dynamics model"

---

> ### Author Response · Authors · 2024-11-26
>
> We sincerely appreciate your recognition of our work, as it means so much to us. We hope the following response can help clarify concerns you may have about our methods:
> * **W1,Q3**: We sincerely appreciate your valuable feedback. We are currently incorporating comparisons with APV, a straightforward yet effective action-free pretraining method based on DreamerV2. To ensure fairness in the comparison, we have implemented APV on the foundation of DreamerV3 to eliminate any confounding factors. This experiment will be included in subsequent revisions of the paper.
> * **W2**: RCWM does have certain limitations. It cannot handle more complex scenarios that may arise in real-world settings, such as a tilted table or viewpoint changes, which could introduce errors in the robot dynamics learned by the world model. Addressing robustness in such cases would require incorporating corresponding disturbances during training. However, this is not the focus of this paper. In future work, we will explore more robust methods for world model learning.
> * **Q1**: Yes, we used the same set of trajectories for evaluation. Specifically, we selected 100 successful trajectories collected by a third-party policy that is independent of the evaluated policies as our evaluation dataset. This ensures fairness. Moreover, successful trajectories contain rich dynamic information, making them highly suitable for evaluating dynamics learning.
> * **Q2**: In the stick-push task, the robot needs to first grasp the stick and then use it to push the kettle. During the stick-grasping phase, the robot may collide with the stick and the table, altering the dynamics and increasing dynamic errors. This highlights a limitation of the RCWM approach: it may struggle to effectively infer the robot's dynamics in scenarios involving significant collisions or resistance. Nevertheless, our method is still more accurate than vanilla world model that uses a single model to learn global dynamics.
> * **Q4**: Thank you very much for your valuable feedback. RCWM is the first step toward decoupling robot dynamics from environmental dynamics. We firmly believe that this decoupling and explicit interaction modeling represent a valuable research direction. Our goal is to enable robots to first learn to predict their own motion and then progressively learn to interact with the external world. However, explicitly modeling interaction relationships can be challenging, especially for complex interactions, such as those involving flexible objects. Currently, RCWM is limited to learning relatively simple interactions, such as pushing objects or opening doors. Furthermore, when factors in real-world scenarios cause changes in robot dynamics, RCWM may struggle to robustly provide accurate dynamic predictions. In future work, we plan to explore methods for learning more robust robot dynamics and more accurately modeling interactions.
>
> Thank you once again for the time and effort you have dedicated to our work!

---

> > ### Comment · Reviewer_tbDh · 2024-11-26
> >
> > Thank you for addressing my comments. I am willing to raise my score to 8 if the authors promise to add the comparison experiments with APV, as well as a detailed discussion section on the limitations of RCWM, including the inability to handle complex scenarios and disturbances, upon publication of this paper.

---

> > > ### Author Response · Authors · 2024-11-27
> > >
> > > We sincerely appreciate your recognition of our method, as it holds great significance for us.
> > >
> > > Due to limited computational resources, the supplementary experiments are still in progress. We promise that we will supplement our comparative experiments with APV and discuss the limitations of RCWM more clearly. Given the impending rebuttal deadline, we will update the experimental results on our website as soon as possible. Please stay tuned.
> > >
> > > Thank you once again for your time and effort in reviewing our work!

---

### Official Review · Reviewer_nEog · 2024-11-02

**Soundness:** 2
**Presentation:** 3
**Contribution:** 2
**Rating:** 5
**Confidence:** 4

**Summary:**

The paper introduces the Robo-Centric World Model (RCWM), designed to improve sample efficiency and robustness in model-based reinforcement learning (MBRL) for visual robot control. RCWM achieves this by decoupling robot dynamics from environment dynamics, allowing each component to be modeled independently while an interaction model, based on cross-attention, captures the effect of robot actions on the environment. The model’s training pipeline includes a mask-guided warmup using robot segmentation masks, followed by mask-free pre-training and fine-tuning, ensuring RCWM can robustly handle novel tasks and disturbances.

**Strengths:**

* The paper is well-written and well-organized.
* The idea of decoupling robot and environment dynamics is very novel and holds potential value for improving the efficiency and robustness of learning algorithms.
* The proposed method demonstrates resilience against visual disturbances and changing backgrounds.

**Weaknesses:**

* More baselines should be included, such as a leading model-based RL algorithm TD-MPC2 (https://arxiv.org/abs/2310.16828). Meanwhile, it is important to report the sample-efficiency comparison with leading model-free algorithms like DrM (https://arxiv.org/pdf/2310.19668).
* More experiments should be conducted in more challenging tasks such as dexterous manipulations in Adroit (https://arxiv.org/pdf/1709.10087) to validate the effectiveness of the proposed method.
* It is also essential to validate the proposed method in real-world experiment (such as Box Close task).

**Questions:**

* How might RCWM handle scenarios where prior robot segmentation masks are unavailable or difficult to obtain? Could an alternative approach to disentangling robot and environment dynamics be feasible in these cases?
* Given RCWM's reliance on cross-attention for modeling interactions, how well would it handle environments with more unpredictable or dynamic elements, such as deformable objects or non-static obstacles?
* What will happen if we only use the replay buffer from 1 task trained by DreamerV3 as the dataset?

---

> ### Author Response · Authors · 2024-11-13
>
> We sincerely thank the reviewers for their valuable comments to make this paper better!
>
> We hope that the following responses will help reviewers better evaluate our approach:
> * **W1**: TD-MPC2 is currently a leading model-based RL algorithm and has garnered significant attention from the community. However, it is primarily focused on locomotion control tasks that rely on state inputs, rather than on visual robot manipulation tasks that depend on image inputs. Additionally, we believe that directly comparing our method with TD-MPC2 would not provide a valid assessment of our approach, as our method is an enhancement based on DreamerV3, and performance differences between TD-MPC2 and DreamerV3 could obscure our method's effectiveness. While our approach could theoretically be adapted to build world models in TD-MPC2, doing so would require substantial modifications. We chose DreamerV3 as the foundation for our algorithm because it demonstrates exceptional potential in visual robot manipulation tasks. Our goal is to investigate whether there exists a more suitable method for constructing world models specifically for visual robot manipulation tasks, while using TD-MPC or Actor-Critic for policy learning simply represents an alternative way to leverage this world model. And for some model-free methods that focus on sample efficiency, the same can be disruptive in evaluating our methods. DrM, for example, uses neural network perturbations to enhance exploration, which is important in Meta-world tasks. There also exist Model-based rl methods for studying efficient exploration that could be combined with our approach to improve exploration efficiency, but this is not the dimension we wish to evaluate. We believe that our comparison with the vanilla world model in DreamerV3 is sufficient to illustrate the effectiveness of our approach and is in the dimension of world model construction.
>
> * **W2**: For dexterous manipulation tasks in Adroit, we believe that state-based control algorithms such as TD-MPC2 are more suitable. While visual model-based rl methods may have difficulty in effectively learning high-dimensional hand dynamics from only images, which remains a current area of endeavor. And the interaction between the dexterous hand and external objects is more complex compared to the gripper, and it is quite difficult to make the world model learn this complex interaction accurately. We believe that our chosen tasks are also challenging, which typically require more than a million interactions to learn effective policies and can sufficiently validate the effectiveness of our approach. To the best of our knowledge, we are the first to separate robot dynamics and environment dynamics and simultaneously model their interactions, which we believe is a valuable and successful attempt. However, for learning more complex robot dynamics and interactions, we will try it in our future work.
>
> * **W3**: RL training in the real world is often costly, and although the method is already sample efficient, it still requires a significant amount of environment interaction. We believe that experiments in simulated environments have sufficiently demonstrated the effectiveness of our approach. We will try real-world experiments in future work if conditions permit.

---

> ### Author Response · Authors · 2024-11-13
>
> We hope that the following responses will help reviewers better evaluate our approach:
>
> * **Q1**: We use SAM2 to obtain robot segmentation masks, a technique that is well established and works well. We describe it in detail in A.4. Only manual labeling of the first frame of each trajectory is required to automatically obtain masks for subsequent frames, which results in lower manual costs. Furthermore, it is important to emphasize that our approach requires only a few data with robot masks to effectively decouple the dynamics of the robot and the environment, and doesn't require additional mask data for subsequent fine-tuning. In the paper, we only used 16 trajectories with a total of 4000 steps with masks for warm-up, which means that even if each trajectory needs to be manually labeled in order to obtain masks using SAM2, only 16 images need to be manually labeled. And further reducing the amount of data the model may still be able to function properly. In fact, even without mask data, our model still exhibits the ability to decouple structured information. We illustrate this in A5.1 as well as in Figure 12. Nevertheless, it is still difficult to stably decouple robot and environment dynamics without prior masks. It is not clear to us if there is a way to do so.
>
> * **Q2**: Learning world models in dynamic stochastic environments is still an open problem. For DreamerV3, its ability to model random parts of the environment, such as random walks of monsters in Atari games, through stochastic representations is still limited. It is quite difficult to accurately model the physical interactions of unpredictable objects. Existing world models are all data-driven in nature and are deficient in modeling dynamic stochastic environments. However, for simple physical interactions such as pushing a cube, opening a door, etc., our approach can effectively learn this interaction and generate realistic imaginary trajectories.
>
> * **Q3**: If only the replay buffer from a single task is used as the dataset for pre-training, it may lead to insufficient learning of the robot dynamics, as the covered state space might not be comprehensive enough. This could prevent the learned robot model from providing accurate dynamic predictions for new tasks. We focus on enabling the same robot to learn a variety of tasks and aim to better utilize the data collected from previous tasks to pre-train the world model. This is useful in real applications.  In fact, as long as the state and action spaces of the robot in the pre-training dataset are sufficiently covered, even data collected through unsupervised exploration rewards in a single task can lead to a good learning of robot dynamics.

---

> ### Author Response · Authors · 2024-11-13
>
> We sincerely hope that the reviewers will re-evaluate our approach in light of our responses. We greatly value any feedback on our responses. Once again, thank you for your valuable suggestions despite your busy schedule!

---

> ### Comment · Reviewer_nEog · 2024-11-25
>
> Thanks for your reply! I have changed my scores to Rating: 3 and Confidence: 3.
>
> The authors did not include any of the experiments I proposed in the original review and responded with incorrect information: TD-MPC2 is **not** a state-based RL algorithm and has been **validated** on diverse manipulation tasks. As a leading model-based RL algorithm, it is reasonable to expect the authors to compare its performance with the proposed method. I hope the authors will include more comparisons in the future.

---

> > ### Author Response · Authors · 2024-11-26
> >
> > we have provided clear reasons why it is unnecessary to compare our method with TD-MPC2, and why our approach is not suited for dexterous hand tasks. However, it seems that you have not carefully read our response. You pointed out that we gave wrong information, but that's just a misunderstanding on your part. I wish to clarify the issue once again:
> >
> > We are certainly aware that TD-MPC2 has a vision-based implementation. However, the original paper does not include any experiments on vision-based manipulation tasks, only experiments on visual DMC. For the dexterous manipulation tasks in Adroit, we believe that state-based inputs are more suitable for MBRL than image-based inputs, as the world model may struggle to effectively learn complex dynamic information. TD-MPC2 demonstrates excellent performance with state-based inputs. **However, this is not relevant to this paper.**
> >
> > We have clarified the relationship between our method and TD-MPC2 in our pinned response. We believe this comparison is not meaningful. It only reflects the performance differences between DreamerV3 and TD-MPC2, rather than addressing the method we proposed. **We emphasize once again that our method focuses on the dimensions of world model construction and pretraining, not policy optimization.** TD-MPC2 and DreamerV3 are fundamentally different in terms of policy learning, which is not the dimension we aim to compare. As for DrM, we see no valid reason to include a comparison with it.  **Our goal is not to stack tricks to improve sample efficiency but to explore a new approach to world model construction and pretraining that is more suitable for robot learning.**  One of the benefits of this approach is the ability to accelerate learning for new tasks. We believe that you may not have fully understood our paper.

---

> > > ### Comment · Reviewer_nEog · 2024-11-26
> > >
> > > Thank you for your detailed clarification! I have updated my scores to Rating: 5 and Confidence: 4.
> > >
> > > Since many visual-based RL algorithms have been successfully tested in the Adroit environment (not just state-based methods), I still believe it would be a valuable complement to the original paper to showcase the capability of the proposed method in world model construction and pretraining in this environment, along with its performance comparison against other MBRL baselines. If the results do not match or over state-of-the-art methods, it would be beneficial to summarize the limitations and propose future directions for improvement.

---

> > > > ### Author Response · Authors · 2024-11-29
> > > >
> > > > We sincerely appreciate the constructive suggestions you have provided.
> > > >
> > > > We acknowledge that our discussion of RCWM's limitations was insufficient, which may have caused some confusion, as pointed out by Reviewer tbDh. The core idea of our method is to extract reusable robot dynamics from previously learned tasks, but there are some limitations to this approach:
> > > > * Firstly, since our input is image, we need to ensure that the tasks in the pretraining data share a consistent viewpoint with the new tasks. Only under this condition can reusable dynamics be effectively extracted. However, in Adroit, the viewpoints are generally inconsistent between tasks, making it nearly impossible to extract reusable dynamics.
> > > > * Secondly, the robot must not be over-obscured. Secondly, the robot should not be excessively occluded. Excessive occlusion can make it difficult for the world model to learn the robot's dynamics. In Adroit, the joints of the dexterous hand often experience severe occlusion, which presents a significant challenge for learning the robot's dynamics.
> > > >
> > > > Our approach is mainly applicable to fixed-viewpoint robot arm visual manipulation tasks. This is a commonly used setup in the field of robotics. However, we do need to recognize that this setup has some limitations. Also the learning of complex robot dynamics such as dexterous hands is yet to be investigated. We will include a comprehensive discussion of the applicability scenarios and limitations of RCWM in the appendix of the paper.
> > > >
> > > > Furthermore, to address your concerns, we will consider adding a comparison with TD-MPC2 in the appendix and discuss the differences and advantages of our method in visual manipulation tasks. This will provide a reference for the community, although we still believe that this comparison is not entirely reasonable. **We will update the comparison results with TD-MPC2 on our website in a timely manner. Please stay tuned**.
> > > >
> > > > We sincerely hope that you will assess our method in its entirety. Our approach aims to enable robots to first learn to predict their own movements, and then gradually learn to predict interactions with the external world. We are the first to realize this idea. This provides a novel perspective for pretraining and constructing world models that are applicable to robotic learning.
> > > >
> > > > We would like to express our sincere gratitude for the time and effort you have devoted to reviewing our work!

---

> ### Author Response · Authors · 2024-12-01
>
> We have updated the comparison results with TD-MPC2 on our [website](https://robo-centric-wm.github.io) and hope you will take a look. Experimentally, our method is competitive with TDMPC2. Since TDMPC2 has significant advantages over DreamerV3 in terms of both policy learning and decision making, our approach focuses on the construction of the world model without modifying the policy learning, and therefore does not show a consistent advantage over TDMPC2. We found that TDMPC2 still has excellent performance even without pre-training. However, since the learning of representations and dynamics relies heavily on task-relevant reward information, it is difficult for TDMPC2 to obtain valid information from pre-training. While our method can efficiently learn representations and robot dynamics that are useful for new task learning. Although our method is not as lightweight as TDMPC2 and requires the use of pre-training, robot masks, etc. to perform well, we believe that task-independent pre-trained world models will be one of the important research directions in the future. And our approach offers a new way of thinking about this.
>
> Once again, we sincerely appreciate the time and patience you have dedicated to reviewing our work.

---

### Official Review · Reviewer_qUFP · 2024-11-03

**Soundness:** 3
**Presentation:** 3
**Contribution:** 2
**Rating:** 3
**Confidence:** 5

**Summary:**

This paper presents robot-centric world models that distinctively model robot and environmental dynamics to enhance visual model-based reinforcement learning. Results from eight MetaWorld tasks demonstrate improved learning efficiency compared to the vanilla baseline, DreamerV3.

**Strengths:**

- The paper is well-organized, featuring visualizations that aid in understanding the proposed robot-centric world models.
- The approach appears to effectively learn robot-centric dynamics, though I have some questions about it.
- Experimental results show performance gains over the state-of-the-art DreamerV3 across 8 MetaWorld tasks.

**Weaknesses:**

- A primary concern is that the performance improvement may not solely be attributed to the proposed idea. Several additional implementations, such as the transformer in the interaction model, mask-guided encoder, and predictor head, confuse the attribution of improvements.
- The generalization ability of the proposed model seems limited. Significant performance drops occur under environmental disturbances, particularly in the Sweep-Into task (Figure 8(b)). Additionally, the focus on disturbance testing should extend to other variables like robot arm appearance, base position, and camera view.
- The selected tasks appear relatively simple. It would be beneficial to evaluate the modeling effectiveness on more complex embodiments, such as dexterous hands or humanoids, as evaluated in [1].
- The absence of ablation studies. An analysis of each design's impact—such as the warm-up stage, warm-up data choice, or removal of components—would assist the understanding.
- Construction error may not be a good metric to measure the ability to model robot dynamics, because the error may mainly caused by the environmental dynamics error. As an extreme case, the vanilla world model may be good at robot dynamics but very poor at environmental modeling. The authors should address this concern.
- Testing the extension capability of the proposed method on other state-of-the-art algorithms, like TD-MPC2 [1], would significantly strengthen this paper.

**Questions:**

- What is the performance in a multi-task setup, following [1]?
- What do the attention maps signify in all figures? The highlighted regions seem meaningless.
- The gripper reconstruction in Figure 5 appears inadequate. Could the authors clarify this?

[1] Hansen, N., Su, H., & Wang, X. TD-MPC2: Scalable, Robust World Models for Continuous Control. In *The Twelfth International Conference on Learning Representations*.

[2] Sferrazza, C., Huang, D. M., Lin, X., Lee, Y., & Abbeel, P. (2024). Humanoidbench: Simulated humanoid benchmark for whole-body locomotion and manipulation. *arXiv preprint arXiv:2403.10506*.

---

> ### Author Response · Authors · 2024-11-13
>
> We sincerely thank the reviewers for their valuable comments to make this paper better!
>
> We hope that the following responses will help reviewers better evaluate our approach:
> * **W1**: We considered the potential performance gains that could result from the use of robot masks, so we added the same mask-guided decoder as RCWM when training DreamerV3 and used the same amount of mask data for pre-training to eliminate the interference. We explain this in line 417 of the paper. For the interaction model, which is an integral part of RCWM, the same is part of the proposed improvements. Our RCWM, by decoupling robot dynamics from environment dynamics, is able to allow us to model the interaction between the two more explicitly, which is one of the advantages of RCWM over the vanilla world model.
>
> * **W2**: Our approach is not an improvement aimed at improving generalization.  The improvement in generalization for environment disturbances is a result of our separation of predictions for robot dynamics and environment dynamics. When the environment changes, our robot branch is able to provide accurate robot representations and dynamics predictions almost unaffected. Thus even if the environment representation changes, the policy can still generate reasonable actions based on the robot representation.  Figure 8 shows that our approach is more robust to environment disturbances compared to DreamerV3, which strongly supports our idea.  As to why both our approach and DreamerV3 produce significant performance degradation on the sweep-into task, we believe that it is because the sweep-into task needs to determine where the holes and objects are by the desktop texture, and thus is very sensitive to the desktop texture. For a description and visualization of the sweep-into task check out https://meta-world.github.io/. Furthermore, since we focus on using the same robot for multiple tasks, there is no need to be robust to robot appearance. And when position and viewpoint change, the dynamics in the 2d image-based world model will be severely affected, which is still lacking research.
>
> * **W3**: The tasks we chose are very challenging tasks in the Meta-world benchmark, which usually require more than a million steps of interaction with the environment to learn effective policy.  And these tasks mostly involve interactions with objects in the environment, which can effectively evaluate whether our method is able to learn such complex interactions.  Our focus is on visual robot manipulation tasks rather than locomotion control tasks. As for the dexterous hand in MyoSuite evaluated in TD-MPC2 or the humanoid robot control task in Humanoidbench, it is very difficult to use visual inputs for control, and the current approach is still attempting on state inputs. Moreover, it is also very difficult to accurately learn the complex dynamics of the dexterous hand from vision, which we believe is beyond the scope of this paper.
>
> * **W4**: Since every component of RCWM is indispensable, we think that ablation experiments are not necessary. For example, without the warm-up phase, it is difficult for our model to stably disentangle representations of the robot and the environment without prior robot masks.  For the choice of warm-up data, which is not critical for this method, it is sufficient that the data covers the operational space relatively adequately and thus allows our model to learn the robot dynamics, or even trajectories collected with unsupervised stochastic exploration. As for the components in RCWM, such as mask-guided decoder, interaction model, etc., RCWM does not work properly after removal.
>
> * **W5**: Although reconstruction error is not a good metric to measure the ability to model robot dynamics, it also reflects the ability to some extent. In order to avoid the influence of environment on evaluating robot dynamics modeling, we multiply the imagined reconstructed image with the ground-truth robot mask and compute the mean square error with the ground-truth robot image to serve as the robot dynamics prediction error. We mentioned in line 447 of the paper. With this evaluation method, we were able to consider only the reconstruction results for the robot part. We perform a full evaluation in Figure 13 in Appendix A.5.2.
>
> * **W6**: TD-MPC2 mainly focuses on locomotion control tasks using state as input rather than images, and does not use reconstruction loss to learn representations. In contrast, our approach focuses on visual robot manipulation tasks and uses reconstruction loss to learn representations. Applying RCWM to TD-MPC2 would require large modifications, which may be beyond the scope of this paper and may lead to new approaches. We believe that the performance improvement over the vanilla world model is sufficient to illustrate the effectiveness of our approach.

---

> ### Author Response · Authors · 2024-11-13
>
> We hope that the following responses will help reviewers better evaluate our approach:
> * **Q1**: Our approach is not an improvement for TD-MPC2, nor does it address the problem of multi-task offline learning. We focus on online training for visual robot manipulation. And, to the best of our knowledge, TD-MPC2 does not perform multi-task learning with images as input either. This remains a difficult problem for the current research and we believe it is beyond the scope of this paper.
> * **Q2**: We detailed the computation of the attention map in A.5.3 and supplemented the visualization results in Figures 15 and 16. The attention map consists of two parts, one for $\hat{z}_t^{R}$ and $z _ {t-1}^E$ with a shape of 32*32 and one for $\hat{h}_t^R$ and $z _ {t-1}^E$ with a shape of 8*32. We concatenate both as 40*32 attention maps and resize to 80*64 to align the size of the image observations for visualization. The parts of the attentional map without significant activation are the attentional maps of $\hat{z}_t^{R}$ and $z _ {t-1}^E$, and the parts showing significant activation are the attentional maps of $\hat{h}_t^R$ and $z _ {t-1}^E$. This is because $\hat{z}_t^{R}$ is a stochastic representation generated by $\hat{h}_t^R$, which contains richer information, such as action information. Although the attention maps for $\hat{z}_t^{R}$ and $z _ {t-1}^E$ are not significant, they are still computed to ensure consistency between the robot and environment representations. The activations are not meaningless either. We observe significant activation when the robot touches an object, suggesting that the interaction model is capable of learning the interaction relationship between the robot and the object.
> * **Q3**: The gripper may lose some details when reconstructed due to its small size, but the opening and closing state of the gripper can also be clearly represented, which does not affect the imagination of the operation. In Figure 5, in the case of occlusion of the gripper, it may appear to disappear due to its small size, but it still does not affect the operation of the door. Methods based on reconstruction usually lose some details when reconstructing, which is due to the nature of the autoencoder. Even though the reconstructed image may be missing some details, the hidden state may not ignore this information. We think reconstruction is just a way to better help us understand what's going on in our world model, because state transitions and decisions happen in the latent space.

---

> ### Author Response · Authors · 2024-11-13
>
> We sincerely hope that the reviewers will re-evaluate our approach in light of our responses. We greatly value any feedback on our responses. Once again, thank you for your valuable suggestions despite your busy schedule!

---

> > ### Comment · Reviewer_qUFP · 2024-11-25
> >
> > Thanks for the response. I genuinely think the robot-centric world model is promising in the future. However, I disagree with some replies and have the following concerns:
> > - **More experiments on other simulation tasks.** MetaWorld is not scene-complex and visually realistic enough to show the potential of RCWM completely. MyoSuite, ManiSkill2, or Robocasa could be evaluated to address this concern, following current MBRL papers on robot manipulation.
> > - **Generalization ability.** The authors did not conduct the experiments to address my proposed concerns. They claimed "there is no need to be robust to robot appearance." and "when position and viewpoint change, the dynamics in the 2d image-based world model will be severely affected, which is still lacking research.". However, these properties are very important to the proposed robot-centric world model, from my perspective. At least, the authors should provide empirical experiments to investigate these.
> > - **TD-MPC2 does have vision-based implementations.** Please check their official code carefully.
> > - **TD-MPC2 mainly focuses on locomotion control tasks using state as input rather than images,...** They focus on greatly many manipulation tasks as well.
> > - **Ablation is very necessary.** The authors claim "ablation experiments are not necessary" by saying "warm-up data is not critical" and "mask-guided decoder, interaction model, etc., RCWM does not work properly after removal." But, how do the authors reach these conclusions? Why are they not important? It would be very useful to provide these studies explicitly.
> > - **Extention to TD-MPC2.** I agree that it is hard to extend the current framework to TD-MPC2, which has no explicit reconstruction loss. However, this is the limitation of this paper. The authors should discuss it and propose how to extend RCWM to TD-MPC2 in the paper. I do not see these changes in the next version of the manuscript.
> >
> > Given that the proposed method does not show additional properties (e.g., generalization ability, multi-task training, etc) compared with SOTA MBRL algorithms such as TD-MPC2 and DreamerV3, the authors should compare RCWM with these methods on diverse tasks to demonstrate their superiority.
> >
> > I wish to see that these concerns will be addressed in the future.

---

> > > ### Author Response · Authors · 2024-11-26
> > >
> > > Thank you sincerely for acknowledging the potential of our approach in the future. I hope my response will sufficiently address your concerns:
> > > * We are very eager to test our method in more simulation environments, but currently, Meta-World is the most suitable for our setting. Firstly, our research focuses on how the same robot can quickly learn a variety of manipulation skills while fully utilizing reusable and useful prior knowledge.  This requires the simulation environment to offer a large number of manipulation tasks. However, ManiSkill2 includes only a limited number of simple tasks such as grasping, which does not align with our setting. RoboCasa, on the other hand, is not RL-friendly; it primarily relies on imitation learning to derive effective policies from datasets. As for MyoSuite, solving such complex, high-dimensional locomotion control tasks with vision-based input is exceedingly challenging. Additionally, our method requires explicit modeling of the interactions between the dexterous hand and the objects, which is highly complex. We believe this represents a promising direction for future research.
> > > * Generalization is indeed crucial for practical applications, but it is an independent research direction and not the primary focus of our method. The generalization capability demonstrated in the paper is merely one of the advantages brought by RCWM. Visual MBRL still lacks sufficient research on viewpoint generalization. As I mentioned, world models based on 2D image inputs are significantly affected by viewpoint changes. It is important to emphasize that this is not the focus of this paper and that it is an area of research that has not been fully explored. While we plan to investigate this in our future work, it is beyond the scope of this paper.
> > > * I am certainly aware that TD-MPC2 has a vision-based implementation. However, the original paper does not include any experiments on vision-based manipulation tasks, only experiments on visual DMC. Our ongoing research is also exploring the potential of TD-MPC2 in vision-based control, but this is unrelated to this paper. I have already detailed the relationship between our method and TD-MPC2 in my pinned response.
> > > * TD-MPC2 does indeed focus on manipulation tasks, but it is based on state inputs rather than images.
> > > * Ablation studies are necessary, but our method cannot be ablated because each component is indispensable. I believe that if you fully understand our model, this question will not arise. As for the claim that pretraining data is not important, this is indeed the case. Our pretraining focuses solely on extracting effective robot dynamics without considering the environment. As long as the pretraining data contains sufficient information about robot dynamics, RCWM can learn reusable robot dynamics effectively.
> > > * Our method focuses on the construction and pretraining of the world model rather than policy optimization. While DreamerV3 and TD-MPC2 differ in world model construction, their approaches are conceptually similar. However, they are fundamentally different in policy learning. I do not believe our method needs to be extended to TD-MPC2. In fact, we could simply modify the policy learning component to a form similar to that of TD-MPC2. However, this has no direct connection to TD-MPC2. Moreover, this comparison pertains to the dimension of policy optimization, whereas our focus is on world model construction.
> > > * I disagree with your statement that "the proposed method does not show additional properties (e.g., generalization ability, multi-task training, etc.) compared with SOTA MBRL algorithms such as TD-MPC2 and DreamerV3."  In fact, we are the first to propose an algorithm that decouples robot dynamics and applies it to new tasks. Our model effectively decouples robot dynamics from environmental dynamics and explicitly models their interaction, which previous models could not achieve. Furthermore, we believe that this decoupling is valuable for future research, as it allows the robot to first learn to control itself and gradually learn to interact with the environment. We believe that the properties of RCWM are distinct and clearly differentiate it from previous methods.
> > >
> > > Thank you for your valuable feedback. However, we still disagree with some of the points you raised. We will work on improving our paper in the future.

---

> ### Comment · Reviewer_qUFP · 2024-11-29
> **Final Response**
>
> Apologies for the delay in response.
>
> I particularly want to thank the authors for taking so much time to explain their settings and answer all the questions I had. Here is a summary of the main concerns that were not addressed by the authors:
>
> - I wish I could make my statement about 'TD-MPC2 should be included as a baseline' clearer. I know that the authors aim to convey a promising direction, which I agree with and appreciate: decoupling robot dynamics and environments is helpful. However, from my perspective, the current submission only verifies the feasibility of this interesting idea, but is far from the conclusion on its generalizable effectiveness. Specifically, the training setup is the same as the one in TD-MPC2 and DreamverV3 but with an additional pre-training phase. In other words, the goal of this manuscript is to improve the accuracy of world modeling. If TD-MPC2 or other SOTA MBRL algorithms, which are not included in this manuscript, can already outperform RCWM by learning from scratch, then it is hard to see the motivation of the community to use the proposed RCWM that even require additional efforts (e.g., SAM 2 annotation) to design the pre-training stage. In this case, using other SOTA MBRL algorithms will be enough and burden-easy.
> - That is also the reason why I wish to see the results on more tasks. Possibly, RCWM may perform badly in tasks with complex interactions. If the authors could show the superiority of RCWM on diverse task domains, even compared with only DreamerV3, the concern above could also be alleviated.
> - That is also the reason why I wish to see additional emergent properties, like generalization ability and robustness to visual disturbance that are not discovered in previous methods. These properties could also greatly motivate the use of RCWM.
>
> Regarding other responses:
>
> - TD-MPC2 is not a solely state-based algorithm, recognized by other reviewers as well. There already exists an extension of TD-MPC2 to visual humanoid control tasks [1], which are likely more complex than manipulation tasks. Based on their results, DreamerV3 significantly underperforms TD-MPC2. Meanwhile, the authors of TD-MPC2 also open-source their code with high reusability to the tasks considered in this manuscript.
> - ManiSkill2 does not include only simple tasks like grasping. Tasks like insertion, stacking, and assembling are also included. I do not think they are simpler than the selected ones. ManiSkill2 also supports more realistic rendering, which may be a better choice to show the promise of RCWM.
> - Regarding MyoSuite tasks or Adroit mentioned by other reviewers, it is true that they are hard to model. Informally speaking, based on my personal experience, these tasks could be successfully learned with DreamerV3 with ~33 PSNR. It would be better to give the conclusion after conducting the results.
> - I also carefully read the discussions with other reviewers. The authors said that "in Adroit, the viewpoints are generally inconsistent between tasks, making it nearly impossible to extract reusable dynamics." I would like to say that the views can simply be consistent across all Adroid tasks, based on my past research. The authors also said that "the joints of the dexterous hand often experience severe occlusion, which presents a significant challenge for learning the robot's dynamics." This is true. But,  Adroid tasks are inherently partially observable and can be solved by DreamerV3 without the additional tuning, based on my research as well. This is not the reason to exclude the possibility of doing experiments in this domain.
> - Ablation studies are necessary. I understand this paper, and I know that some components cannot be removed. What I wish to see are some other design choices, like model size, pre-training dataset, model choice, etc. Doing more experiments on design choices will help to guide the community to extend RCWM in the future. Meanwhile, the authors tell me some conclusions (in both manuscript and rebuttal page), but do not show the numerical results to verify them. I would suggest providing more empirical numbers to strengthen some words.
>
> In summary, I would maintain my score and genuinely suggest the authors address the concerns above with more empirical results.
>
> [1] Hansen, N., SV, J., Sobal, V., LeCun, Y., Wang, X., & Su, H. (2024). Hierarchical World Models as Visual Whole-Body Humanoid Controllers. arXiv preprint arXiv:2405.18418.

---

> > ### Author Response · Authors · 2024-12-01
> >
> > We sincerely appreciate the valuable time and patience you have dedicated to us, as well as the insightful feedback you provided. We fully agree with the issues you raised, and we will incorporate your suggestions to improve our paper in the future.
> >
> > We have added a comparison experiment with TD-MPC2 on our [website](https://robo-centric-wm.github.io/), and we hope you will take a look. Although TD-MPC2 performs similarly and is more lightweight and simpler than our method, we still believe that task-agnostic pre-trained world models are a valuable research focus for the future.
> >
> > Finally, once again, our sincerest respect for all that you have done for us!

---

### Official Review · Reviewer_pYBH · 2024-11-06

**Soundness:** 2
**Presentation:** 3
**Contribution:** 2
**Rating:** 6
**Confidence:** 4

**Summary:**

This paper considers the problem of world model pre-training in robotics applications. Different from previous approaches that use a single model to train the scene evolution, the key idea of this paper is to decouple world dynamics into robot dynamics and environment dynamics. The proposed model uses a dedicated robot branch to predict the future images of the robot, an environment branch to predict the future images of the environment, and an interaction module to inject the information from the robot branch to the environment branch.

**Strengths:**

This paper is well-written and the logic is clear. The idea of decupling robot dynamics and environment dynamics is interesting. The claim is supported by promising experiment results.

**Weaknesses:**

It is unclear if the gains are from the additional capacity introduced by the additional branch. (2x model capacity?) To verify this, the authors could expand the dreamer model capacity and compare the results.

It is strange that the robot branch can learn effects of the environment on the robot, even there is no information exchange from environment branch to the robot branch.
This makes me wonder if the true technical idea is simply using two models to predict the evolution of the robot and the rest of the scenes separately. This can be validated by disabling the interaction part and compare with the proposed model in terms of dynamics prediction and policy learning.

**Questions:**

See my questions above

---

> ### Author Response · Authors · 2024-11-20
>
> We sincerely appreciate your recognition of our work, as it means so much to us.  We hope the following response can help clarify concerns you may have about our methods:
>
> * **W1**: To eliminate interference caused by model capacity, we reduced the latent space dimension when constructing the two branches in RCWM. The latent space dimension of the dynamics model in DreamerV3 is 512, with 5.7M parameters, whereas the latent space dimension of the dynamics model in RCWM is 256, with a total of 3.7M parameters for both branches and the interaction model combined. In fact, our method uses fewer parameters in the dynamics model. This also indirectly demonstrates that RCWM is an efficient approach to constructing world models. We believe that expanding the capacity would likely result in performance improvements, as validated in DreamerV3.
> * **W2**: Our aim is to learn generalized robot dynamics that can be directly used for new tasks through pre-training, therefore hoping that it can inference independently of the environment. This prevents us from designing the information exchange between branches, even though doing so might lead to better performance.  We explicitly models the effect of the robot on the environment while implicitly modeling the effect of the environment on the robot.  If the interaction model is disabled, the environment branch will not be able to access the action information and the details of the robot's movement and thus it will not work properly. If we simply use two identically constructed branches to predict the dynamics of the robot and the environment separately, we believe that this may not allow for effective dynamic separation and consistent interaction prediction. Since the learning of policies requires multi-step rollout in the world model, if there is no information exchange between two branches completely, the prediction results of the two branches may produce significant differences during rollout due to the presence of compounding errors thus failing to produce consistent interaction prediction. We therefore believe that constructing an interaction model is necessary, and this is the key to the success of our approach in modeling the manipulation process.
>
> Thank you once again for the time and effort you have dedicated to our work!

---

> > ### Comment · Reviewer_pYBH · 2024-11-28
> >
> > I would like to thank the authors for the responses. The response addressed my first question. The responses suggested what will happen if the information exchange is disabled, but it would be nice if this can be included in the baseline since it is a key design factor. I would like to maintain my score of 6.

---

> ### Comment · Area_Chair_SfiK · 2024-11-25
> **Please read rebuttal**
>
> Dear Reviewer pYBH, Could you please read the authors' rebuttal and give them feedback at your earliest convenience? Thanks. AC

---

### Author Response · Authors · 2024-11-21

## **We hereby summarize some common misunderstandings and concerns raised by the reviewers regarding our work.**

First, we would like to emphasize that our work is oriented towards robot manipulation tasks with purely visual inputs. Our approach is improved based on DreamerV3, since the Dreamer series of methods has been widely used to solve visual robot manipulation tasks.

* **The reviewers mentioned the comparison with TD-MPC2 and the expansion based on it, and we would like to clarify the difference and comparability of our method with TD-MPC2**. The main contribution of our approach is to improve the construction of the world model in Dreamerv3 to make it more suitable for learning robot manipulation tasks. Although DreamerV3 and TD-MPC2 are both advanced model-based algorithms, there are significant differences between them. First, there are significant differences in the construction of the world models. DreamerV3 uses reconstruction loss to learn the hidden space representation and constructs RSSM to learn the dynamics. In contrast, TD-MPC2 uses contrast learning to learn the hidden space and uses MLP to learn dynamics with frame stacking as input. Second, the two use completely different approaches for policy learning. Moreover, TD-MPC2 focuses more on control tasks with state as input rather than visual control tasks, while DreamerV3 focuses more on visual control tasks. Therefore, we believe that the comparison with TD-MPC2 does not reflect the improvement of our approach because our improvement focuses on the world model, while the two are quite different both in terms of world modeling and policy learning. Moreover, there are obvious problems in extending our approach to TD-MPC2 due to the different construction forms of the world model. **We would like to clarify that our approach and TD-MPC2 are not directly comparable due to fundamental differences in world modeling and policy learning.**.
* The reviewers refer to evaluations on more complex control tasks, such as dexterous hand or humanoid robot control tasks. We would like to emphasize that our approach makes a valuable attempt to effectively decouple robot dynamics as well as interaction learning. However, learning complex robot dynamics such as dexterous hands from visual input is still an open problem. The fact that dexterous hands have many joints and are often occluded when performing manipulation tasks which leads to the difficulty of learning such high-dimensional dynamics simply from 2D images. Moreover, the interaction between dexterous hands and objects is much more complex compared to gripper-object interaction, and trying to learn this interaction accurately may require more complex models and more data. Although our approach may not be effective in solving such complex tasks, we believe that we have performed a valuable exploration and provided a new way of thinking about world model construction. Learning about complex robot dynamics is our future endeavor.

---

### Meta-Review · Area_Chair_SfiK · 2024-12-17

**Metareview:**

This paper proposes a world-model-based framework for robot learning. However, the paper does not have enough experiments and ablative studies to prove the effectiveness of the proposed idea. Moreover, the generalization ability is not tested in the paper. MIssing baselines give another issue. Overall, the paper is below the bar of ICLR.

**Additional Comments On Reviewer Discussion:**

The authors addressed some of the reviewers' concerns. However, the core challenges about baselines, ablation studies, and span of benchmarks are not fully resolved.

---

### Decision · Program_Chairs · 2025-01-22

Reject